# Physiological regulation of neuronal Wnt activity is essential for TDP-43 localization and function

Nan Zhang [1,4], Anna Westerhaus[1,4], Macey Wilson [1,3], Ethan Wang [1], Loyal Goff[1,2] & Shanthini Sockanathan [1✉]

## Abstract

**Nuclear exclusion of the RNA- and DNA-binding protein TDP-43 can induce neurodegeneration in different diseases. Diverse processes have been implicated to influence TDP-43 mislocalization, including disrupted nucleocytoplasmic transport (NCT); however, the physiological pathways that normally ensure TDP-43 nuclear localization are unclear. The six-transmembrane enzyme glycerophosphodiester phosphodiesterase 2 (GDE2 or GDPD5) cleaves the glycosylphosphatidylinositol (GPI) anchor that tethers some proteins to the membrane. Here we show that GDE2 maintains TDP-43 nuclear localization by regulating the dynamics of canonical Wnt signaling. Ablation of GDE2 causes aberrantly sustained Wnt activation in adult neurons, which is sufficient to cause NCT deficits, nuclear pore abnormalities, and TDP-43 nuclear exclusion. Disruption of GDE2 coincides with TDP-43 abnormalities in postmortem tissue from patients with amyotrophic lateral sclerosis (ALS). Further, GDE2 deficits are evident in human neural cell models of ALS, which display erroneous Wnt activation that, when inhibited, increases mRNA levels of genes regulated by TDP-43. Our study identifies GDE2 as a critical physiological regulator of Wnt signaling in adult neurons and highlights Wnt pathway activation as an unappreciated mechanism contributing to nucleocytoplasmic transport and TDP-43 abnormalities in disease.**

**Keywords** GDE2; Neurodegeneration; Nucleocytoplasmic Transport; TDP-43; WNT
**Subject Categories** Membranes & Trafficking; Molecular Biology of Disease; Neuroscience

## Introduction

Abnormalities in the RNA/DNA-binding protein TDP-43 (transactive response DNA-binding protein of 43 kD) are associated with neurodegeneration across multiple neurodegenerative diseases that include amyotrophic lateral sclerosis (ALS), ALS-frontotemporal dementia (FTD), and Alzheimer's disease (AD) (Kim and Taylor, 2017; Ling et al, 2013; Tziortzouda et al, 2021). Hallmarks of TDP-43 pathologies comprise nuclear exclusion and cytoplasmic accumulation, which ultimately result in TDP-43 aggregation and downregulation of expression that disrupt its functions in RNA metabolism, mRNA transport, and stress granule formation (Ling et al, 2013; Sun et al, 2017a; Tziortzouda et al, 2021). Multiple neuropathologies converge to compromise TDP-43 nuclear localization, including cytoplasmic protein aggregates that can sequester TDP-43 in the cytoplasm, impaired nucleocytoplasmic transport (NCT), and alterations in the nuclear pore complex (NPC) (Kim and Taylor, 2017; Moore et al, 2020; Winton et al, 2008). However, little is known about the physiological mechanisms that normally maintain nuclear TDP-43 expression and if these pathways are compromised in disease. Bridging this knowledge gap will provide insight into fundamental cellular pathways that regulate TDP-43 biology and clarify mechanisms that might contribute to familial and sporadic disease pathologies.

The six-transmembrane protein Glycerophosphodiester phosphodiesterase 2 (GDE2 or GDPD5) functions at the cell surface to cleave the glycosylphosphatidylinositol (GPI)-anchor that tethers some proteins to the membrane (Park et al, 2013; Rao and Sockanathan, 2005; Yanaka, 2007). During development, GDE2 expression in neurons regulates the production of select spinal and cortical neuronal subtypes and controls oligodendrocyte development by coordinating the release of soluble maturation factors (Choi et al, 2020; Rodriguez et al, 2012; Sabharwal et al, 2011). Recent studies identify unique roles for GDE2 in the adult nervous system that are separate from its developmental functions and suggest potential involvement in regulating neuronal survival and function (Cave et al, 2017; Nakamura et al, 2021; Westerhaus et al, 2022). GDE2 positively regulates ADAM10 (A Disintegrin and Metalloproteinase-Domain containing protein 10) α-secretase processing of amyloid precursor protein (APP) to generate neuroprotective sAPPα and occlude production of toxic Aβ42 peptides through cleavage-dependent inhibition of the GPI-anchored metalloprotease inhibitor RECK (Reversion inducing Cysteine-rich protein with Kazal motifs) (Nakamura et al, 2021). Loss of GDE2 reduces sAPPα amounts, increases Aβ peptides, and

[1]The Solomon Snyder Department of Neuroscience, The Johns Hopkins School of Medicine, 725 N Wolfe Street, Baltimore, MD 21205, USA. [2]McKusick-Nathans Department of Genetic Medicine, Kavli Neurodiscovery Institute, The Johns Hopkins School of Medicine, 725 N Wolfe Street, Baltimore, MD 21205, USA. [3]Present address: Department of Cellular Biology, University of Georgia, Biological Sciences 302, 120 Cedar St., Athens, GA 30602, USA. [4]These authors contributed equally: Nan Zhang, Anna Westerhaus. ✉E-mail: ssockan1@jhmi.edu

accelerates plaque deposition in the *APPSwe* mouse model of amyloidosis (Nakamura et al, 2021). Moreover, mice lacking GDE2 show behavioral abnormalities, including deficits in short and long-term spatial memory observed in mouse models of AD and ALS/FTD and motor abnormalities observed in models of ALS (Cave et al, 2017; Daudelin et al, 2024). Notably, GDE2 accumulates intracellularly in AD patient neurons, and RECK release is impaired in AD postmortem brain, suggesting that the erosion of GDE2 physiological function may contribute to amyloid pathologies and behavioral changes in AD (Nakamura et al, 2021).

GDE2 intracellular accumulation and impaired GPI-anchored protein release are also detected in ALS and ALS/FTD (Westerhaus et al, 2022), where TDP-43 pathologies are prominent, but the role of GDE2 in ALS and ALS/FTD disease pathology is unknown. Here, we show that mice lacking GDE2 (*Gde2*KO) recapitulate the progression of TDP-43 nuclear exclusion and downregulation in disease and that these changes correlate with deficits in NCT and NPC abnormalities. GDE2 loss causes the persistent activation of canonical Wnt signaling in neurons, which is sufficient to cause the disruption of NCT, the NPC, and TDP-43 distribution and function across mouse and human models. Strikingly, cells with aberrant GDE2 intracellular accumulation in ALS postmortem brain correlate closely with TDP-43 mislocalization and downregulation. Induced human spinal neuron (iSN) models of ALS carrying the GGGGCC hexanucleotide repeat expansion in the *C9orf72* gene (C9ALS) display GDE2 abnormalities, TDP-43 dysfunction, and erroneous Wnt activation that, when inhibited, partially rescues TDP-43 molecular function. Taken together, our findings suggest that GDE2 normally acts to constrain canonical Wnt signaling in neurons to preserve appropriate NCT and TDP-43 nuclear expression and that GDE2 failure and consequent Wnt activation may contribute to TDP-43 nuclear exclusion and downregulation in disease.

# Results

## GDE2 is required for appropriate TDP-43 expression

We first determined if the loss of GDE2 function results in abnormalities in TDP-43 expression by examining TDP-43 distribution by immunohistochemistry in aged WT and *Gde2*KO mice. TDP-43 expression was markedly reduced in sections of 19-month-old *Gde2*KO cortex compared with WT controls (Fig. 1A–E). Consistent with TDP-43 loss of function, bulk RNAseq from 19-month cortical tissue identified 84 transcripts with splicing errors in *Gde2*KOs but not in WT animals (FDR < 0.1) and 48 significantly altered transcripts (FDR < 0.1, |Δ percent spliced-in (PSI)| > 0.1). Motif discovery analysis of the significant alternatively spliced introns showed an enrichment of the consensus TDP-43 binding site consisting of the TG repeat motif, with 43 mis-spliced transcripts harboring the TG repeat motif (Fig. 1F; Dataset EV1; Appendix Fig. S1) (Lukavsky et al, 2013). Validating the disruption of TDP-43 RNA-splicing function in *Gde2*KO animals, RT-PCR of the neuronal TDP-43 target *Camk1g* (Jeong et al, 2017; Ling et al, 2015) showed increased cryptic exon incorporation in 19-month *Gde2*KOs compared with WT (Fig. 1G,H). *Camk1g* is expressed in neurons, astrocytes, and oligodendrocyte precursor cells in mice (Zhang et al, 2014); accordingly, this increase is commensurate with decreased neuronal

TDP-43 expression in *Gde2*KO animals (Fig. 1E). Cryptic exons were also detected in two other TDP-43 targets, *Tecpr1* and *Synj2bp* (Sinha et al, 2024), in *Gde2*KOs at 13 months (Fig. 1I) although this varied between animals. This may be due to the stochastic downregulation of TDP-43 in different neuronal subtypes across animals. Nevertheless, all *Gde2*KOs tested showed evidence of TDP-43 dysfunction. TDP-43 ablation or knockdown results in neurodegeneration across several models, including mice and iSNs (Vanden Broeck et al, 2014). Accordingly, we examined aged *Gde2*KO animals for hallmarks of neurodegeneration by immuno-histochemistry. Cortical sections of 19-month animals showed a robust increase in astrogliosis (GFAP) and changes in microglia coverage and morphology consistent with microglial activation (Iba1) in *Gde2*KOs compared with WT (Fig. EV1A-d''',I,J), in line with increased inflammation. Moreover, *Gde2*KO animals display neuronal loss, particularly in deep-layer neurons where GDE2 expression is enriched (Daudelin et al, 2024; Yao et al, 2021) (Fig. EV1E–H,K,L). Thus, GDE2 is required for the appropriate expression and function of TDP-43 and for cortical neuronal survival.

## GDE2 function in adults is required for appropriate TDP-43 expression

GDE2 has established roles in developmental neurogenesis and in oligodendrocyte maturation (Choi et al, 2020; Park et al, 2013; Rao and Sockanathan, 2005; Rodriguez et al, 2012; Sabharwal et al, 2011). To distinguish whether GDE2 regulation of TDP-43 expression and neuronal survival involves its developmental or adult expression, we genetically ablated GDE2 at 2 months of age by administering tamoxifen to *R26Cre-ER;Gde2fl/-* mice. Animals were aged to 13 months and compared to age-matched tamoxifen injected *R26Cre-ER;Gde2 + /−* controls. This inducible strategy retains GDE2 developmental activity in *R26Cre-ER;Gde2fl/-* mice but ablates its function in the adult. Tamoxifen treatment resulted in an ~60% reduction of GDE2 protein levels in *R26Cre-ER;Gde2fl/-* mice compared with *R26Cre-ER;Gde2 + /−* controls (Appendix Fig. S2A). TDP-43 expression was reduced in tamoxifen injected *R26Cre-ER;Gde2fl/-* mice at 13 months (Fig. 1J–L), and this reduction was similar to TDP-43 decreases observed in 13-month *Gde2*KOs (Fig. 1M; Appendix Fig. S2B). Further, 13-month-old tamoxifen injected *R26Cre-ER;Gde2fl/-* mice showed evidence of neurodegeneration that recapitulated neurodegenerative changes in global 13-month *Gde2*KO animals, specifically robust microglial activation (Appendix Fig. S2C,D,K,O) with no significant changes in astrogliosis or neuronal numbers compared to controls (Appendix Fig. S2E–J,L–N,P–R). Taken together, these observations indicate that GDE2 expression in the adult is required for TDP-43 expression and to prevent neurodegenerative changes.

## Gde2KOs show age-progressive abnormalities in TDP-43 distribution

Previous studies show that forced cytoplasmic expression of exogenous human TDP-43 in transgenic mice downregulates endogenous TDP-43 expression (Igaz et al, 2011; Winton et al, 2008), suggesting that the mislocalization of TDP-43 to the cytoplasm is one initiator of TDP-43 downregulation and nuclear exclusion in disease. To determine if TDP-43 is mis-localized in

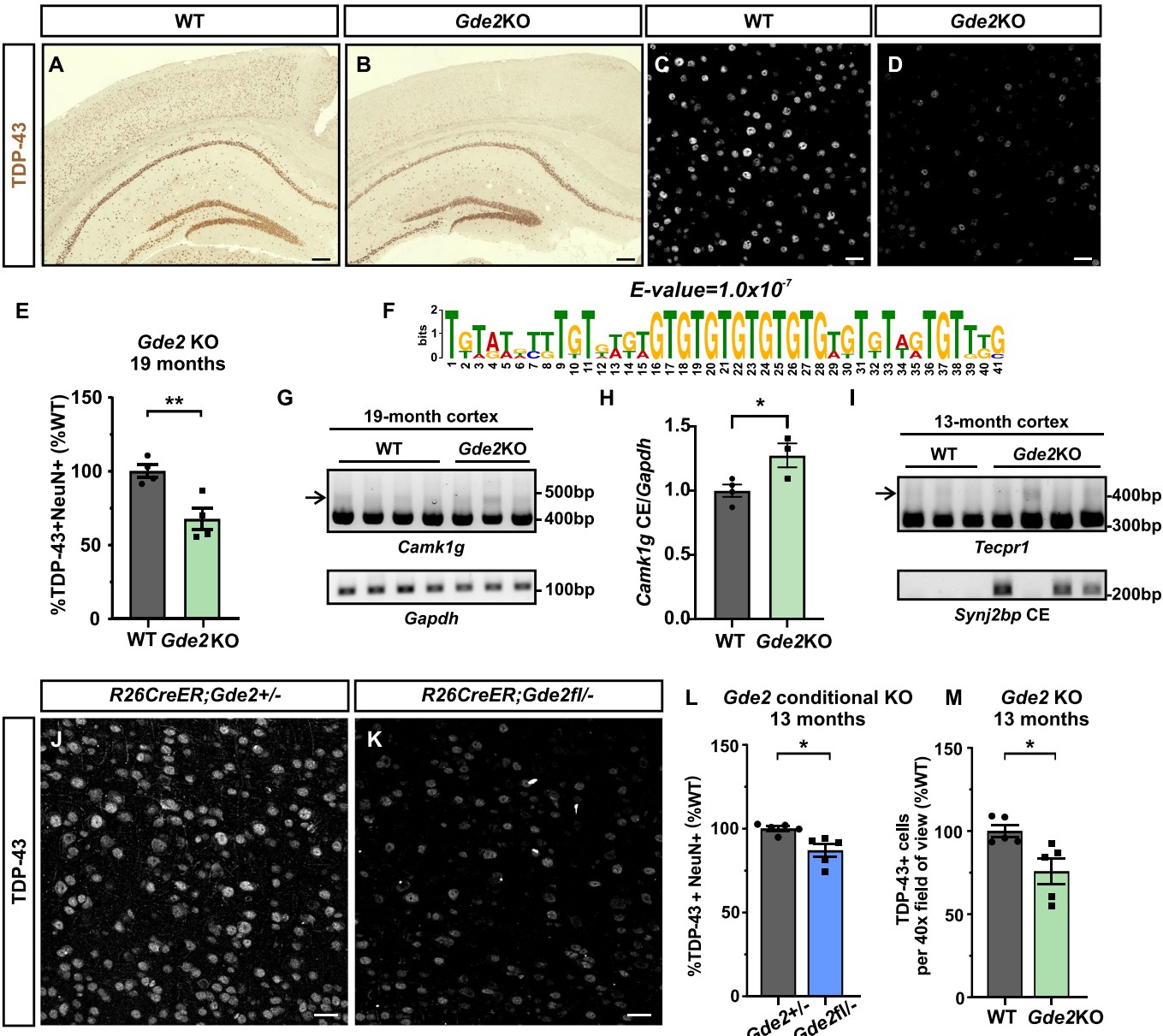

**Figure 1. GDE2 loss results in TDP-43 downregulation.**

(A–D) Representative images of immunohistochemical and immunofluorescence staining for TDP-43 in 19-month WT and *Gde2*KO parietal cortex. (E) Quantification of the percentage of neurons with TDP-43 expression. **$P = 0.0084$, $n = 4$ animals/genotype. (F) Enrichment of TDP-43-binding motif from RNAseq of *Gde2*KO 19-month cortex. *E*-value $= 1.0 \times 10^{-7}$, $n = 3$ animals/genotype. (G, H) RT-PCR and quantification of cryptic exon incorporation in *Camk1g* (arrow) in 19-month WT and *Gde2*KO cortices. *$P = 0.0359$, $n = 4$ WT, 3 *Gde2*KO. (I) RT-PCR of cryptic exon incorporation in *Tecpr1* (arrow) and *Synj2bp* in 13-month WT and *Gde2*KO cortices. (J, K) Representative images of TDP-43 immunohistochemical staining of cortical sections of 13-month conditional *Gde2*KO animals (*R26Cre-ER;Gde2fl/-*) where GDE2 function is genetically ablated at 2 months by tamoxifen delivery. *R26Cre-ER;Gde2 +/−* animals injected with tamoxifen at 2 months are controls. (L, M) Graphs quantifying the percentage of neurons with TDP-43 expression in 13-month controls and *Gde2* conditional KOs (L, *$P = 0.0132$), and WT and global *Gde2*KOs (M, *$P = 0.0196$) $n = 5$ animals/genotype. All graphs: mean ± s.e.m. Unpaired *t* test: (E, H, L, M). Scale bar: (A, B) $= 100$ μm; (C, D, J, K) $= 50$ μm. Source data are available online for this figure.

*Gde2*KO animals, we examined WT and *Gde2*KO animals at earlier timepoints, tracking backwards to 7 months and 4 months of age. 7-month *Gde2*KO animals showed a reduction in the number of neurons expressing TDP-43 in addition to reduced TDP-43 nuclear intensity in TDP-43 expressing neurons (Fig. EV2A–D,G,H). At 4 months, the number of neurons that expressed TDP-43 was equivalent between WT and *Gde2*KO animals (Fig. 2A,B,K).

However, TDP-43 expression was increased in the cytoplasm in *Gde2*KO neurons compared with WT, although nuclear intensities were unchanged (Figs. 2A-b",L, and EV2J). Thus, GDE2 ablation results in age-progressive changes in TDP-43 localization, beginning with increased cytoplasmic accumulation, followed by nuclear loss and subsequent downregulation of expression (Fig. EV2L). These observations suggest that TDP-43 cytoplasmic localization at

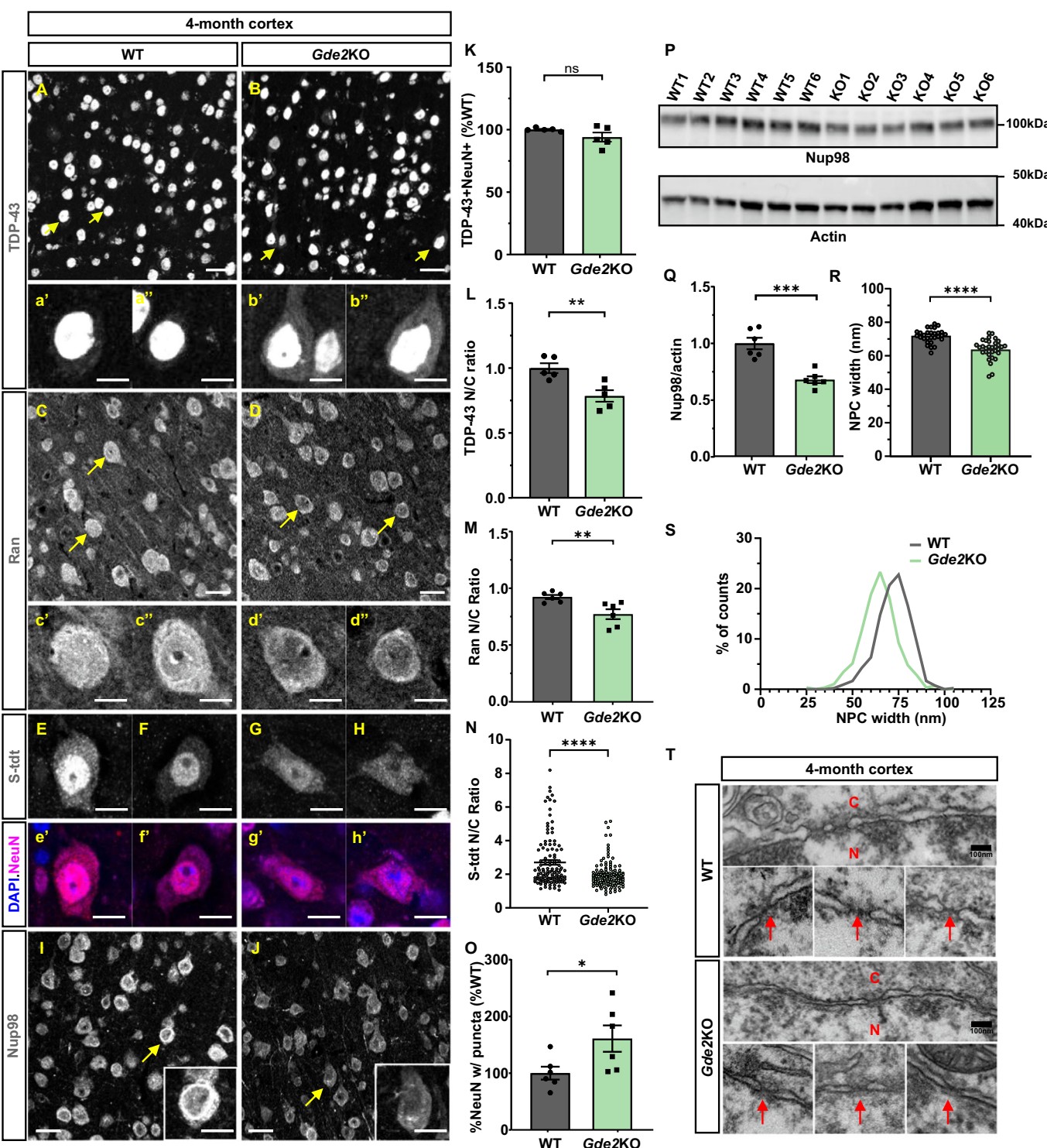

4 months is a critical initiator of TDP-43 downregulation in aged *Gde2*KO animals.

## Gde2KOs show NCT and NPC deficits

NCT defects are associated with TDP-43 mislocalization (Chou et al, 2018; Coyne et al, 2021; Hutten and Dormann, 2020; Kim and Taylor, 2017; Moore et al, 2020; Zhang et al, 2015). The Ras-related GTPase (Ran) controls the nuclear import and export of proteins through a gradient of nuclear-enriched Ran-GTP compared with cytoplasmic Ran-GDP (Moore and Blobel, 1994). Nuclear import of TDP-43 utilizes the Ran gradient (Moore et al, 2020; Pinarbasi et al, 2018; Tziortzouda et al, 2021), which can be visualized by the ratio of nuclear versus cytoplasmic Ran expression (N/C). To determine

**Figure 2. *Gde*2KOs show deficits in NCT and NPC.**

(A–J) Representative images of immunostained cortical sections. Arrows highlight examples at higher magnification showing TDP-43 cytoplasmic localization (b', b"), Ran nuclear reduction (d', d"), S-tdt nuclear reduction (G, H) and Nup98 puncta and downregulation (inset, J) in *Gde*2KO neurons. (K–O) Quantification of *Gde*2KOs compared with WT (K) TDP-43+ neuronal numbers (ns $P = 0.1791$, $n = 5$ WT, 5 *Gde*2KO, >250 neurons per animal), (L) TDP-43 N/C ratio (**$P = 0.0063$, $n = 5$ WT, 5 *Gde*2KO, >150 neurons per animal), (M) Ran N/C ratio (**$P = 0.0096$, $n = 6$ WT, 6 *Gde*2KO, >150 neurons per animal), (N) S-tdt N/C ratio (****$P = 3.03134 \times 10^{-6}$, $n = 118$ WT cells, 129 *Gde*2KO cells, 4 animals/genotype), (O) percent neurons with Nup98 puncta (*$P = 0.0405$, $n = 6$ WT, 6 *Gde*2KO, >400 neurons per animal). (P) Western blot of protein extracts from 4-month-old animals normalized to actin, with quantification shown in (Q) (***$P = 0.0003$, $n = 6$). (R–T) TEM analysis of NPC width (****$P = 1.6990 \times 10^{-8}$, $n = 29$ WT cells, 33 *Gde*2KO cells, 3 animals/genotype) (R, S) with exemplar micrographs in (T). C cytoplasm, N nucleus. Arrows highlight NPCs. All graphs: mean ± s.e.m. Welch's t test: (K, N, R); unpaired t test: (L, M, O, Q); Scale bar: (A–D, I, J) = 25 µm; (a'–b", c'–d", E-H, e'–h', I (inset), J (inset)) = 10 µm; (T) = 100 nm. Source data are available online for this figure.

if NCT is disrupted in 4-month-old *Gde*2KOs at the time of TDP-43 mislocalization, we quantified the Ran gradient in WT and *Gde*2KO cortical neurons. At 4 months, *Gde*2KO neurons show reduced Ran N/C ratios compared with WT, suggesting that NCT is impaired in the absence of GDE2 (Fig. 2C-d",M). The Ran N/C ratio is also decreased at 7 months when TDP-43 nuclear intensity is reduced, consistent with continued disruption of NCT (Fig. EV2E,F,I). As an independent measure of NCT, we generated an adeno-associated virus (AAV) shuttle construct expressing a tdtomato reporter containing nuclear localization (NLS) and nuclear export (NES) signals (S-tdt) (Zhang et al, 2015), and delivered virus into 1-month-old *Gde*2KO and WT mice. Analysis of 4-month-old animals showed a 28% reduction in the N/C ratio of tdtomato localization in *Gde*2KO cortical neurons compared with WT (Fig. 2E–h',N), providing support that GDE2 loss results in disrupted NCT.

NCT deficits are linked to NPC abnormalities such as down-regulation, aberrant distribution, and cytoplasmic accumulation of NPC proteins (Hutten and Dormann, 2020; Kim and Taylor, 2017; Moore et al, 2020). The NPC consists of approximately 30 different nucleoporin (Nup) proteins and F/G (Phenylalanine-Glycine) Nups interact with transport proteins to regulate NCT (Hutten and Dormann, 2020; Kim and Taylor, 2017; Moore et al, 2020). Nup98 is required for the appropriate assembly of the NPC and the selective nuclear import of proteins (Wu et al, 2001). Further, aberrant Nup98 cytoplasmic localization and aggregates are observed in neurodegenerative diseases such as ALS, ALS/FTD, and AD (Coyne et al, 2020; Eftekharzadeh et al, 2018; Hutten and Dormann, 2020; Moore et al, 2020). Four-month *Gde*2KO animals showed an increase in Nup98 cytoplasmic puncta compared with WT, accompanied by a decrease in Nup98 staining (Fig. 2I,J,O). Western blot of cortical extracts showed a reduction in Nup98 protein in 4-month *Gde*2KO mice, confirming Nup98 down-regulation (Fig. 2P,Q). The large number of NPC proteins precludes their comprehensive examination; accordingly, we utilized transmission electron microscopy (TEM) to gain a global view of NPC ultrastructure in *Gde*2KO neurons at 4 months of age. TEM revealed a decrease in NPC width but not numbers (Figs. 2R–T and EV2K) in *Gde*2KO cortical neurons compared with WT, supporting abnormalities in NPC structure and function that likely contribute to NCT deficits when GDE2 is ablated.

## Wnt activation correlates with TDP-43 changes

To identify GDE2-dependent pathways that impact NCT and TDP-43 localization in neurons, we performed bulk RNAseq from dissected 4-month *Gde*2KO and WT cortices. 82 genes were differentially expressed in *Gde*2KOs (effect size b > 0.5, FDR < 0.1, see Dataset EV2) and of these, Gene set enrichment analysis (GSEA) identified several candidate pathways, including the Wnt pathway (Appendix Fig. S3A; Dataset EV2). Changes in Wnt signaling are associated with disease progression in AD (Bai et al, 2020; Palomer et al, 2019) and abnormalities in Wnt activation are prevalent in spinal neurons (iSNs) derived from induced pluripotent stem cells (iPSCs) from ALS and ALS/FTD patients (Hawkins et al, 2022). Both neuroprotective and neurotoxic roles for Wnts have been reported, with a recent study showing that elevated Wnt signaling in C9ALS iSNs results in reduced numbers of synaptic proteins (Chen et al, 2023); however, the contribution of Wnt activation to NPC, NCT, and TDP-43 mislocalization has not been tested.

During canonical Wnt signaling, Wnt-receptor binding causes the Wnt co-receptor LRP6 to be phosphorylated, which sequesters and prevents GSK3-β from phosphorylating β-catenin and targeting it for degradation (Freese et al, 2010; Rim et al, 2022). Stabilized β-catenin traffics to the nucleus through Ran-independent mechanisms where it associates with TCF/LEF1 to regulate transcription (Rim et al, 2022). Neuronal GDE2 activates Wnt signaling in early postnatal animals to coordinate oligodendrocyte maturation and axonal myelination (Choi et al, 2020). To determine whether GDE2 regulates Wnt pathways in adult neurons, we utilized an established genetic reporter where the H2B-MYC-GFP fusion reporter gene under the control of TCF/LEF1 binding sites is integrated into the *Rosa26* genetic locus (*Wnt-MYC-GFP*; see Methods) (Cho et al, 2017). This strategy enables canonical Wnt activation to be monitored in vivo at single-cell resolution by nuclear MYC or GFP expression. In 4-month-old *WT;Wnt-MYC-GFP* animals, robust MYC expression was detected in non-neuronal cells that comprised oligodendroglia and astrocytes, with some expression in neurons (Fig. 3A-a",C; Appendix Fig. S3B–M). In contrast, *Gde*2KO;*Wnt-MYC-GFP* mice showed a marked increase in the percentage of nuclear MYC+ neurons (Fig. 3B–b",C), while the numbers of MYC+ non-neuronal cells were unchanged (Appendix Fig. S3N). Confirming neuronal Wnt activation, the amounts of β-catenin and activated β-catenin detected by western blot and immunocytochemistry were markedly increased in cultured *Gde*2KO cortical neurons compared with WT (Fig. 3D,E; Appendix Fig. S4). These observations indicate that GDE2 ablation leads to the abnormal activation of canonical Wnt signaling in adult neurons. Of note, neurons with nuclear MYC expression in *Gde*2KO;*Wnt-MYC-GFP* mice showed TDP-43 cytoplasmic localization and decreased TDP-43 N/C ratios in contrast to neurons that lack nuclear MYC expression in *Gde*2KO;*Wnt-MYC-GFP* or *WT;Wnt-MYC-GFP* animals

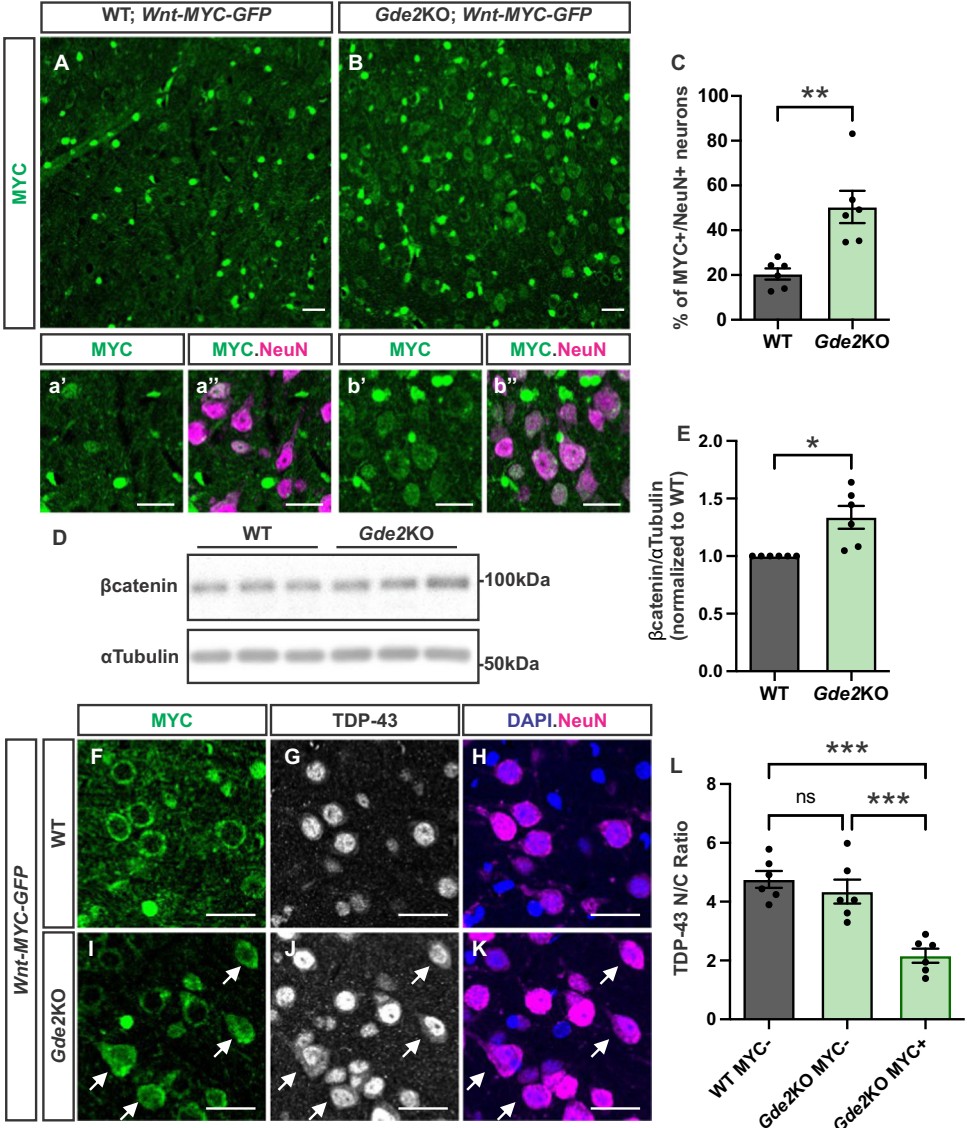

**Figure 3. GDE2 loss results in Wnt activation in neurons.**

(**A–b″**) Representative images of immunostained cortical sections for MYC and NeuN in 4-month-old WT and *Gde2*KO *Wnt-MYC-GFP* mice. (**C**) Graph quantifying the percentage of neurons expressing the MYC reporter in cortical sections. **P = 0.0029, n = 6 animals/genotype. >600 cells per animal. (**D**) Western blots of DIV (days in vitro) 21 primary cortical neuron protein extracts. (**E**) Graph quantifying β-catenin protein level normalized to α-tubulin. *P = 0.0189, n = 5 cultures. (**F–K**) Representative images of immunostained 4-month-old cortical sections for MYC, TDP-43, and NeuN. (**L**) Graph quantifying TDP-43 N/C ratios comparing MYC- neurons in WT and *Gde2*KOs (ns P = 0.6338), MYC- neurons in WT and MYC+ neurons in *Gde2*KOs (***P = 0.0001), and MYC- neurons and MYC+ neurons in *Gde2*KOs (***P = 0.0006). n = 6 animals/genotype, >60 cells per condition. All graphs: mean ± s.e.m. Unpaired *t* test (**C**); one sample t and Wilcoxon test (**E**); one-way ANOVA with Tukey's multiple comparisons test (**L**). Scale bar: (**A, B, a′–b″, F–K**) = 25 µm. Source data are available online for this figure.

(Fig. 3F–L). Thus, neurons with aberrant Wnt activation in *Gde2*KO mice correlate precisely with TDP-43 cytoplasmic accumulation.

## Stabilized β-catenin causes NCT and TDP-43 deficits

Transient activation of Wnt signaling is required for appropriate neuronal development and function (Freese et al, 2010; Mulligan and Cheyette, 2012). To test if sustained neuronal Wnt activation is causal for NPC and NCT defects that result in TDP-43 cytoplasmic

accumulation, we utilized Cre-lox recombination to stabilize endogenous β-catenin in WT neurons in vivo. *Ctnnb*^flex3 mice have engineered lox-P sites flanking Exon 3 of β-catenin, which encodes the sites for GSK3-β phosphorylation that target β-catenin for degradation (Harada et al, 1999). Removal of Exon 3 by Cre-mediated recombination stabilizes endogenous β-catenin, which increases its nuclear entry and subsequent activation of Wnt target genes. We delivered AAV-Cre.GFP and AAV-GFP by retro-orbital injection of virus particles in *Ctnnb*^flex3 mice at P28 and examined animals 3 weeks later for NCT, NPC distribution, and TDP-43

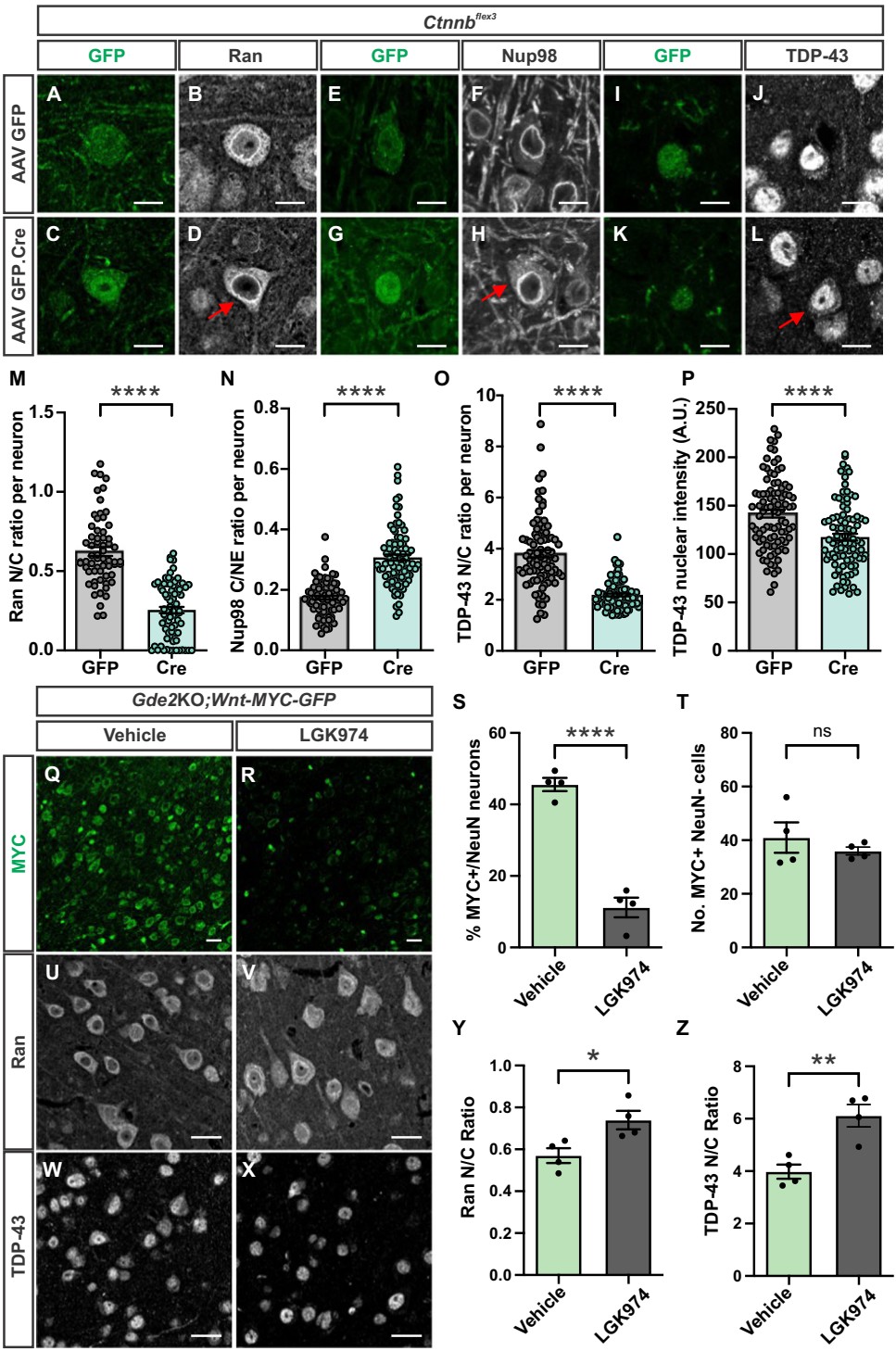

localization in cortical neurons. Neurons expressing Cre.GFP showed a 60% decrease in the Ran N/C ratio (Figs. 4A–D,M and EV3D) and a twofold increase in cytoplasmic accumulation of Nup98 compared to neurons expressing GFP (Figs. 4E–H,N and EV3C). Further, Cre.GFP-expressing neurons showed a concomitant decrease in TDP-43 nuclear intensities and N/C ratio relative to neurons expressing GFP alone (Figs. 4I–L,O,P and EV3A,B). Thus, stabilizing endogenous β-

catenin is sufficient to disrupt NCT and NPC protein distribution and promote TDP-43 nuclear exclusion and cytoplasmic accumulation.

## Wnt inhibition rescues NCT/TDP-43 changes

We next examined if downregulating Wnt signaling in *Gde2*KO animals rescues NCT and TDP-43 cytoplasmic localization.

**Figure 4. Wnt activation drives NPC, NCT, and TDP-43 abnormalities.**

(A–L) Representative images of immunostained cortical sections of AAV-transduced Ctnnb$^{flex3}$ animals 3 weeks post-injection. Arrows highlight Ran nuclear exclusion (D), Nup98 (H), and TDP-43 (L) cytoplasmic accumulation. (M–P) Graphs quantifying neurons expressing GFP or Cre recombinase comparing (M) Ran N/C ratio (****$P = 3.28 \times 10^{-19}$, $n = 63$ GFP, 76 Cre-expressing cells, 3 animals), (N) Nup98 C/Nuclear envelope (NE) ratio (****$P = 2.51 \times 10^{-21}$, $n = 83$ GFP, 91 Cre-expressing cells, 3 animals), (O) TDP-43 N/C ratio (****$P = 1.17 \times 10^{-18}$, $n = 91$ GFP, 94 Cre-expressing cells, 3 animals) and (P) TDP-43 nuclear intensity (****$P = 8.47 \times 10^{-6}$, $n = 91$ GFP, 94 Cre-expressing cells, 3 animals). (Q, R) Representative images of immunostained cortical sections of 4-month-old Gde2KO;Wnt-MYC-GFP animals treated with vehicle or LGK974 for MYC. (S, T) Graphs quantifying percent MYC+ neurons (****$P = 4.71 \times 10^{-5}$) (S), and non-neuronal cells (ns $P = 0.4263$) (T) in cortical sections of 4-month-old animals, $n = 4$ vehicle-treated, 4 LGK974-treated animals. (U–X) Immunostaining of cortical sections of Gde2KO;Wnt-MYC-GFP 4-month-old animals treated with vehicle or LGK974 for Ran (U, V) and TDP-43 (W, X). (Y, Z) Graphs quantifying Ran N/C ratio (*$P = 0.0243$) (Y), and TDP-43 N/C ratio (**$P = 0.0055$) (Z) in cortical sections of 4-month-old animals, $n = 4$ vehicle-treated, 4 LGK974-treated animals. All graphs: mean ± s.e.m. Welch's $t$ test: (M-P); unpaired $t$ test: (S, T, Y, Z). Scale bar: (A–L) = 10 μm; (Q, R, U–X) = 25 μm. Source data are available online for this figure.

LGK974 is an established small-molecule inhibitor of Porcupine, which is required for the palmitoylation and secretion of all Wnt ligands (Liu et al, 2013). LGK974 effectively downregulates canonical Wnt activation at picomole concentrations, is well tolerated in vivo in multiple animal models, and is in Phase I clinical trials for select cancers (Jiang et al, 2013; Liu et al, 2013; Zhang and Lum, 2016). Treatment of Gde2KO cultured cortical neurons with LGK974 decreased the amount of non-phosphorylated β-catenin as assayed by Western blot, indicating that LGK974 is capable of dampening Wnt activation in Gde2KO neurons in vitro (Fig. EV3E,F). We next delivered LGK974 by daily intraperitoneal (IP) injection to 3.5-month Gde2KO;Wnt-MYC-GFP animals when TDP-43 abnormalities were already detected. Analysis of 4-month Gde2KO;Wnt-MYC-GFP mice showed that LGK974 treatment downregulated nuclear MYC expression in neurons compared to vehicle-treated animals, confirming effective dampening of abnormal neuronal Wnt signaling (Fig. 4Q–S but did not affect MYC expression in non-neuronal cells (Fig. 4Q,R,T). Notably, LGK974-treated animals showed a robust increase in the Ran N/C ratio, suggesting that NCT deficits are rescued (Fig. 4U,V,Y), and this was accompanied by an increase in the TDP-43 N/C ratio, implying restored nuclear localization of TDP-43 (Fig. 4W,X,Z). These observations suggest that neuronal Wnt signaling causes NCT and TDP-43 abnormalities in Gde2KO neurons and that pharmacological downregulation of Wnt secretion can reverse these phenotypes in vivo.

## GDE2 and TDP-43 abnormalities overlap in ALS

Our studies so far indicate that GDE2 is an essential physiological regulator of neuronal Wnt activation and that unchecked Wnt signaling in neurons leads to NCT deficits and TDP-43 nuclear exclusion. To determine if GDE2 disruption contributes to TDP-43 abnormalities in disease, we took advantage of earlier observations that GDE2 aberrantly accumulates in intracellular compartments of patients with AD, ALS, and ALS/FTD (GDE2$^{acc}$) (Nakamura et al, 2021; Westerhaus et al, 2022). We co-stained sections of postmortem motor cortex with published, validated antibodies specific to human GDE2 (Nakamura et al, 2021) and TDP-43 focusing on tissue from patients with ALS because of the prevalence of TDP-43 pathologies in ALS and tissue availability (see Appendix Table S1 for demographic information). Confirming published observations (Westerhaus et al, 2022), patients with ALS showed a robust increase in the number of GDE2$^{acc}$ cells (see "Methods") compared with controls (Figs. 5A–D and EV4A). GDE2$^{acc}$ cells in both sets of patients display TDP-43 abnormalities that include

nuclear exclusion, cytoplasmic accumulation, and downregulation of expression, and in some cases, phosphorylated TDP-43 (Figs. 5A–D and EV4A–D,G–J). In contrast, cells with nuclear TDP-43 expression showed little to no GDE2 accumulations in controls and patients with ALS (Figs. 5A–D and EV4E,F). The coincidence of cells with GDE2 accumulations and TDP-43 abnormalities suggests that the disruption of GDE2 physiological function may contribute to TDP-43 disease pathology.

To determine if GDE2 disruption, Wnt activation, and TDP-43 abnormalities coincide in disease, we turned to a human model of ALS. Hexanucleotide GGGGCC repeat expansions in C9orf72 (C9ALS) are causative of familial and sporadic ALS and ALS/FTD (Taylor et al, 2016), and patients with C9ALS show TDP-43 pathologies that include TDP-43 nuclear exclusion and splicing abnormalities (Ma et al, 2022; Tziortzouda et al, 2021; Vanden Broeck et al, 2014). We generated iSNs from iPSCs of control individuals (Ctrl) and patients with C9ALS through an established direct differentiation protocol that generates iSNs by day 18 (Coyne et al, 2020; Workman et al, 2023) (Fig. EV4K–O). In this model, overall levels and distribution of TDP-43 are unchanged but TDP-43 molecular dysfunction is recapitulated by the reduction of specific TDP-43 target gene mRNAs by day 46 of culture (Coyne et al, 2021; Klim et al, 2019; Ma et al, 2022; Melamed et al, 2019; Prudencio et al, 2020). Of the genes previously shown to be regulated by TDP-43, we observed a 40% reduction of Stathmin2 (STMN2) and ELAVL3 in C9ALS iSNs compared with Ctrl (Fig. 5E) and confirmed earlier observations that the distribution of TDP-43 is unchanged (Fig. EV4P). Preceding TDP-43 functional loss, western blot of protein extracts showed that C9ALS iSNs had increased non-phosphorylated β-catenin protein relative to Ctrl iSNs (Fig. 5F,G). Notably, C9ALS iSNs also showed a reduction in GDE2 protein levels, in line with reduced GDE2 activity (Fig. 5H,I). Thus, GDE2 impairments, Wnt activation, and TDP-43 abnormalities are correlated in C9ALS iSNs.

## Wnt activation causes TDP-43 dysfunction

To test if sustained Wnt activation is sufficient to induce TDP-43 dysfunction in iSNs, we treated Ctrl iSNs with an established pharmacological inhibitor of GSK3-β kinase activity, SB216763 (Coghlan et al, 2000), from days 25 to 32. Treatment with SB216763 did not overtly alter the morphology of iSNs or the expression of neuronal markers such as ISLET1 (Workman et al, 2023), which is expressed by postmitotic motor neurons and dorsal spinal interneurons (Fig. EV5A–C,M,N) or increase stress granule formation or autophagy (Fig. EV5D–L,O,P). Treatment of Ctrl lines

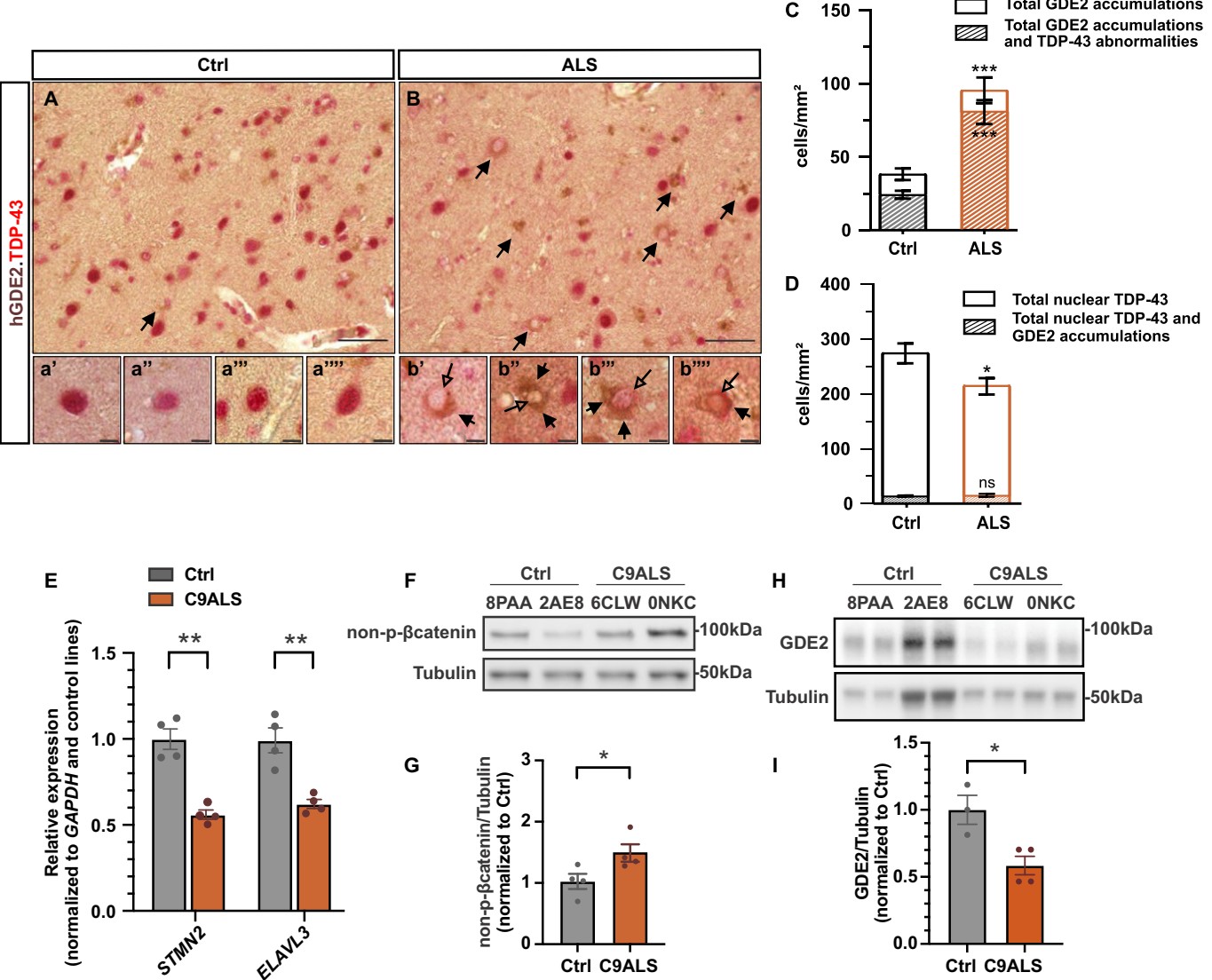

**Figure 5. GDE2 and TDP-43 abnormalities overlap in ALS.**

(A, B) Immunohistochemical staining of postmortem human motor cortex sections. Arrows in (A, B) highlight neurons with GDE2 accumulations and TDP-43 mislocalization or loss. Subpanels (a'–a''''), highlight neurons with nuclear TDP-43 and normal expression of membrane GDE2; subpanels (b'–b'''') show neurons with GDE2 accumulations (arrows) and TDP-43 abnormalities (open arrows) in ALS. (C) Quantification shows an increase in cells with GDE2 accumulations (***$P$ = 0.0002) and GDE2 accumulations colocalized with TDP-43 abnormalities (***$P$ = 0.0002) in ALS compared with control. (D) Quantification of cells with GDE2 accumulation and TDP-43 nuclear expression in control and ALS (ns $P$ = 0.6542). Cells with nuclear TDP-43 expression are reduced in ALS (*$P$ = 0.0144). (C, D) $n$ = 5 Control, 6 ALS. (E) Graph quantifying mRNA amounts of TDP-43 targets by qPCR in Ctrl and C9ALS iSNs at day 46, *STMN2* **$q$ = 0.002246, *ELAVL3* **$q$ = 0.005122. $n$ = 4 Ctrl and 4 C9ALS iSN lines. (F, G) Representative western blot of protein extracts prepared from day 18 (F) and day 32 (H) Ctrl and C9ALS iSNs. (G, I) Graphs quantifying the amount of non-phosphorylated β-catenin *$P$ = 0.0495 (G) and GDE2 *$P$ = 0.0186 (I). $n$ = 3 or 4 Ctrl and 4 C9ALS iSN lines. All graphs: mean ± s.e.m. Welch's $t$ test: (C); unpaired $t$ test: (D, G, I); multiple $t$ test with Welch correction, multiple comparisons correction by FDR (two-stage linear step-up procedure of Benjamini, Krieger and Yuketieli): (E) Scale bar: (A, B) = 50 μm; (a'–a'''', b'–b'''') = 15 μm. Source data are available online for this figure.

with SB216763 resulted in a dose-dependent increase of non-phosphorylated β-catenin, suggestive of Wnt activation (Fig. 6A,B). Of note, iSNs derived from different iPSC lines showed varied sensitivity to SB216763 treatment (Fig. 6A,B). SB216763 treatment did not alter the total amounts of TDP-43 (Fig. 6A,C) in line with published observations showing that TDP-43 overall amounts are not reduced in C9ALS iSNs (Coyne et al, 2021; Coyne et al, 2020). However, nuclear expression of TDP-43 was reduced in iSNs treated with SB216763 (Fig. 6E,F). To test if the nuclear function of

TDP-43 was impaired following SB216763 treatment, we examined a panel of TDP-43 target mRNAs known to be downregulated in response to TDP-43 loss of function (Coyne et al, 2021; Klim et al, 2019). Analysis of SB216763 treated iSNs by qPCR showed a dose-dependent reduction of 6 established TDP-43 target mRNAs (Fig. 6D). These changes were selective and not due to global downregulation of mRNAs as levels of two neuronal markers, *ISL1* and *TUBB3* (Workman et al, 2023), and a panel of housekeeping genes are not altered in response to SB216763 (Fig. EV5Q). In line

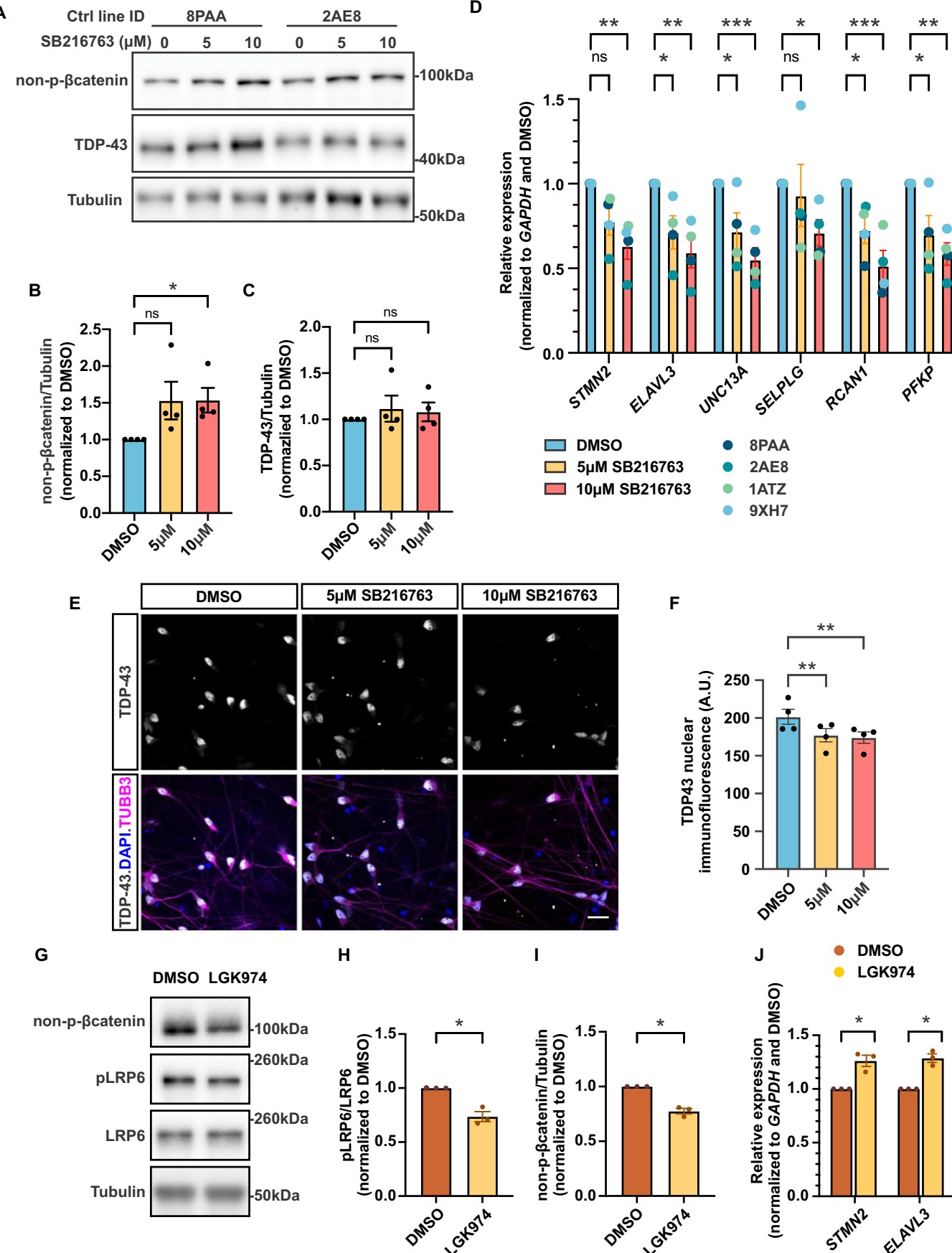

◄

**Figure 6. Wnt activation is causal for TDP-43 dysfunction in iSNs.**

(A) Representative western blot of protein extracts prepared from day 32 Ctrl iSNs treated with vehicle (DMSO) or the GSK3-β inhibitor, SB216763, for 1 week. (B, C) graphs quantifying non-phosphorylated β-catenin normalized to tubulin (B) when treated with DMSO and 5 μM SB216763 (ns $P = 0.1542$) or 10 μM SB216763 (*$P = 0.0267$), and TDP-43 normalized to α-tubulin (C) when treated with DMSO and 5 μM SB216763 (ns $P > 0.9999$) or 10 μM SB216763 (ns $P > 0.9999$). $n = 4$ Ctrl iSN lines. (D) Graph quantifying mRNA amounts of TDP-43 targets by qPCR in Ctrl iSNs treated with DMSO, 5 μM or 10 μM SB216763. Individual lines are color-coded. *STMN2* (DMSO/5 μM) ns $P = 0.0995$, (DMSO/10 μM) **$P = 0.0043$; *ELAVL3* (DMSO/5 μM) *$P = 0.0287$, (DMSO/10 μM) **$P = 0.0016$; *UNC13A* (DMSO/5 μM) *$P = 0.0323$, (DMSO/10 μM) ***$P = 0.0005$; *SELPLG* (DMSO/5 μM) ns $P = 0.7706$, (DMSO/10 μM) *$P = 0.0272$; *RCAN1* (DMSO/5 μM) *$P = 0.0372$, (DMSO/10 μM) ***$P = 0.0002$; *PFKP* (DMSO/5 μM) *$P = 0.0216$, (DMSO/10 μM) **$P = 0.0012$. $n = 4$ Ctrl iSN lines, 2 differentiations/line. (E) Immunostaining of day 32 Ctrl iSNs treated with DMSO or SB216763 for 1 week. (F) Graph quantifying nuclear fluorescence intensity of TDP-43 in TUBB3+ cells when treated with DMSO and 5 μM SB216763 (**$P = 0.0088$) or 10 μM SB216763 (**$P = 0.0050$). $n = 4$ Ctrl iSN lines. (G–J) Representative Western blot of C9ALS iSNs treated with DMSO or LGK974, with graphs quantifying the amount of normalized pLRP6/LRP6 (*$P = 0.0292$) (H), non-phosphorylated β-catenin (*$P = 0.0127$) (I) and qPCR of *STMN2* (*$q = 0.039838$) and *ELAVL3* (*$q = 0.039335$) mRNAs (J). $n = 3$ C9ALS iSN lines, 2 or 3 differentiations/line. All graphs: mean ± s.e.m. Friedman's test with Dunn's multiple comparisons test: (B, C); two-way ANOVA with Dunnett's multiple comparisons test: (D); RM one-way ANOVA with Dunnett's multiple comparisons test: (F) one sample $t$ and Wilcoxon test: (H, I); multiple $t$ test with Welch correction, multiple comparisons correction by FDR (two-stage linear step-up procedure of Benjamini, Krieger, and Yuketieli): (J). Scale bar: (E) = 20 μm. Source data are available online for this figure.

with published observations in iSNs with modest changes in TDP-43 localization (Held et al, 2023), we did not detect cryptic exon inclusion in TDP-43 target transcripts (Fig. EV5R). Taken together, these observations suggest that β-catenin stabilization and accordingly, Wnt activation is sufficient to impair TDP-43 nuclear expression and function in human iSNs.

### Wnt inhibition increases expression of TDP-43 targets

To determine if inhibiting abnormal Wnt activation in C9ALS iSNs mitigates TDP-43 nuclear dysfunction, we treated C9ALS iSNs with the Porcupine inhibitor LGK974 from days 39 to 46. Western blot of protein extracts prepared from C9ALS iSNs treated with LGK974 showed reduced phosphorylation of the Wnt receptor LRP6 and reduced amount of non-phosphorylated β-catenin compared with C9ALS lines treated with vehicle alone, indicating the suppression of Wnt activation (Fig. 6G–I). To determine the consequences of dampened Wnt signaling on TDP-43 nuclear function in C9ALS iSNs, we analyzed *STMN2* and *ELAVL3* mRNAs by qPCR in C9ALS lines treated with vehicle and LGK974. *STMN2* and *ELAVL3* mRNA levels were increased by ~25% in LGK974-treated C9ALS iSNs compared with iSNs treated with vehicle alone (Fig. 6J) indicating a partial rescue of TDP-43 function (Figs. 5E and 6J). Of the four lines tested, iSNs from CS8KT3 were refractory to LGK974 treatment, with no Wnt suppression and limited rescue of TDP-43 target expression (Fig. EV5S–U), suggesting varied sensitivity among different patient cell lines. Taken together, these observations suggest that dampening neuronal Wnt activation mitigates TDP-43 nuclear deficiencies in select C9ALS patient contexts.

## Discussion

We show here that GDE2 encodes a physiological pathway that inhibits the sustained activation of canonical Wnt signaling in adult neurons. Failure to downregulate persistent neuronal Wnt activation causes the erosion of NPC integrity, the disruption of Ran-dependent NCT, and ultimately, TDP-43 mislocalization and downregulation, which results in defective TDP-43 RNA-splicing functions and consequent neurodegeneration. Abnormal GDE2 protein distribution coincides with TDP-43 abnormalities in postmortem tissue of patients with ALS and with Wnt activation and TDP-43 dysfunction in iSN models of C9ALS, and functional studies in C9ALS iSNs provide evidence that Wnt activation is causal for TDP-43 loss of function. These observations suggest that the

disruption of GDE2 physiological regulation of Wnt signaling in neurons may contribute to Wnt-dependent NPC, NCT, and TDP-43 pathologies in disease. Previous studies outline contrasting neuroprotective and neurodegenerative roles for the Wnt pathway (Bai et al, 2020; Hawkins et al, 2022; Palomer et al, 2019). Our findings introduce a new lens to interpret the complexity of Wnt signaling during disease progression to suggest that sustained Wnt activation in neurons is causal for neurodegeneration and is a likely indicator of worsening disease.

The Wnt pathway plays key roles in neuronal development and function that include axon guidance, dendritic arborization, neuron/glial communication, and synaptic biology (Ciani and Salinas, 2005; Freese et al, 2010). Transient activation of Wnt signaling in neurons is critical for these functions, and this is consistent with our observations that the *Wnt-MYC-GFP* reporter is activated in a small number of neurons in WT animals. In contrast, many neurons in *Gde2*KO animals show Wnt activation that coincides with TDP-43 abnormalities, suggesting that persistent Wnt activation is detrimental to neuronal health. This is supported by our observation that forced, prolonged activation of the canonical Wnt pathway in mouse and human iSN models is sufficient to elicit disruptions in NPC, NCT, and TDP-43 localization and function. This underscores the importance of physiological pathways such as GDE2 that precisely regulate the dynamics of Wnt activation in neurons to maintain neuronal viability and function in the adult nervous system. Indeed, while our studies here show that GDE2 negatively regulates Wnt activation in adult neurons, developmentally, GDE2 is required to sustain Wnt signaling in early postnatal neurons (Choi et al, 2020). How GDE2 differentially regulates Wnt signaling in different contexts warrants further investigation. GDE2 function relies on its GPI-anchor cleaving activity, raising the possibility that GDE2 inhibits canonical Wnt activation in adult neurons by cleaving and inactivating surface GPI-anchored mediators of Wnt signaling. Established GPI-anchored substrates of GDE2, such as RECK and members of the heparan sulfate proteoglycan Glypican family that include GPC6 and GPC4 (Cho et al, 2017; Matas-Rico et al, 2016; Park et al, 2013), are known regulators of Wnt activation (Cho et al, 2017; Han et al, 2005; Lebensohn et al, 2016) highlighting them as candidates for further investigation. Moreover, it will be of interest to determine if other physiological regulators of Wnt signaling converge to prevent the neurodegenerative consequences of sustained Wnt activation in neurons throughout life; however, these pathways remain to be elucidated.

Historically, the role of Wnt signaling in neurodegenerative disease to date has been perplexing due partly to contrasting observations from studies of postmortem tissues. For example, proteomics studies using patient tissue reveal that Wnt signaling components are increased during AD progression (Bai et al, 2020), while other studies show a reduction of LRP6 and β-catenin mRNA and protein in AD brains (Liu et al, 2014). Nevertheless, a major caveat is the difficulty in assessing the activation state of Wnt pathways in these contexts. The use of iSNs from patient iPSCs has helped address some of these difficulties, showing that Wnt signaling is elevated in C9ALS iSNs and in iSNs containing ALS-associated mutations in FUS (fused in sarcoma), and contributes to decreased synaptic protein numbers and cell death respectively (Chen et al, 2023; Hawkins et al, 2022). Our findings in C9ALS iSNs combined with our studies in mice and human postmortem tissue identify the disruption of GDE2 activity as a potential mechanism that may contribute to Wnt activation in disease and further suggest that sustained Wnt activation is an upstream driver of NCT/NPC abnormalities and consequent TDP-43 dysfunction. GDE2 abnormally accumulates in intracellular compartments in ALS, ALS/FTD and it is likely that the reduction of GDE2 surface expression underlies its failure in the disease (Nakamura et al, 2021; Westerhaus et al, 2022). Developmental studies show that GDE2 trafficking to the plasma membrane is dependent on thiol-redox regulation of reactive cysteine residues located in the extracellular enzymatic domain and that high oxidative conditions prevent GDE2 surface expression (Yan et al, 2015). While further work is needed to define how GDE2 trafficking is disrupted in disease, we speculate that increased oxidative states in aging and disease may be factors in GDE2 intracellular sequestration.

LGK974 treatment is sufficient to reverse NCT and TDP-43 abnormalities in mouse models and partially rescue TDP-43 nuclear function in C9ALS iSNs, highlighting the therapeutic potential for inhibiting Wnt pathway components to mitigate these cellular abnormalities in disease. However, one limitation is that not all C9ALS iSNs tested were sensitive to LGK974 treatment despite showing Wnt activation. This observation suggests that potential therapies targeting the Wnt pathway may require dose-response studies and/or the development and use of combinations of novel small-molecule inhibitors of Wnt activation or, alternatively, that some cases would be refractory to treatment. Accordingly, it will be important to determine if aberrant Wnt activation is observed in other contexts of TDP-43 dysfunction, including but not limited to iSNs from patients with familial and sporadic forms of ALS, and to test the efficacy of Wnt inhibition in mitigating TDP-43 splicing deficiencies. One implication from our work is that nuanced manipulation of Wnt signaling should be considered in therapeutic development. Our observations suggest that clinical trials that may result in chronic Wnt activation should be carefully designed to avoid potential detrimental effects on neuronal viability and function that could accelerate neurodegeneration.

# Methods

## Study design

The objectives of this study were to (i) evaluate the role of GDE2 in maintaining TDP-43 localization and expression, and neuronal health and survival in the brain, (ii) determine the contribution of the canonical Wnt signaling pathway to the mechanism by which GDE2 maintains TDP-43 localization, and (iii) evaluate the effects of abnormal Wnt signaling activation in disease using human models of *C9orf72* ALS. For studies using patient postmortem tissue, immunohistochemical analyses were performed on the motor cortex of 5 controls and 6 patients; regions of interest (ROIs) for each section were randomly selected for analyses and quantification. For studies using mouse tissues, the number of animals and cells analyzed per animal are noted in corresponding figure legends. Neuronal cell culture experiments were performed with 4–6 biological replicates and three technical replicates per condition. No statistical power analysis was used to predetermine sample size. Sample sizes (number of animals and cells analyzed) were determined to be similar or exceed those previously reported in the literature. For experiments using iPSC lines, 4 nonneurological control and 4 *C9orf72* lines were used in all experiments, with at least 2 independent differentiations for each cell line. For Wnt inhibitor treatment of ALS iSNs, one cell line (CS8KT3) was excluded from the analysis for functional rescue as the cell lines exhibit no response in Wnt suppression. The results from this cell line are included in Fig. EV5. No other data were excluded from the study. For quantitative image analysis, investigators were blinded to genotypic information except in cases where the analysis was automated. iPSCs were obtained via the Answer ALS Project via the Cedar Sinai iPSC repository and tested for pluripotency by the Cedars-Sinai iPSC Core. iSNs were validated with known neuronal markers previously reported using the same differentiation protocol. No mycoplasma contamination was detected throughout the course of the study. Human postmortem tissues were obtained from the Johns Hopkins ALS Postmortem Tissue Core. The use of both human postmortem tissue and iPSCs were approved by the Johns Hopkins Institutional Review Board (IRB). Antibodies were either purchased commercially or extensively validated in previous studies.

## Animals

Mice were bred and maintained in accordance with approved Johns Hopkins University Institutional Animal Care and Use Committee protocols. *Gde2-/-* (Sabharwal et al, 2011), *RosaCreER;Gde2^{lox/−}* (Cave et al, 2017), *Rosa26 Tcf/Lef H2B-EGFP-6xMYC* (Cho et al, 2017), and *Ctnnb1^{flex3}* (Harada et al, 1999) mice were bred, maintained and genotyped as described previously. Male and female mice were used for analysis.

Tamoxifen (Sigma-Aldrich) was emulsified in sunflower oil (Sigma-Aldrich) at 20 mg/ml and delivered to *RosaCreER;Gde2^{lox/−}* mice by 3 subsequent daily intraperitoneal (i.p.) injections (75 mg/kg) beginning at p60. For experiments with Wnt inhibitor treatment in *Gde2-/-;Rosa26 Tcf/Lef H2B-EGFP-6xMYC* mice, LGK974 (Selleck) was reconstituted at 50 mg/mL in N,N-Dimethylacetamide (DMA), aliquoted for daily injections and stored in glass vials at −80 °C. LGK974 was dissolved in 5% DMA in sunflower seed oil and delivered to mice via i.p. injections at 25 mg/kg for 2 weeks, 6 days per week, beginning at 3.5 months old.

## AAV vector construction and injections

All AAV plasmid backbones were based on AAV-GFP/Cre (Kaspar et al, 2002) (Addgene, 49056). AAV-GFP was constructed by deleting the coding sequence of Cre. The coding sequence for

S-tdtomato was synthesized from Lentiviral-S-tdtomato (Zhang et al, 2015) (Addgene, 112579) and subcloned into the AAV backbone by replacing GFP/Cre. The AAV vectors were packaged by Janelia Viral Tools using the PHP.eB capsid (Chan et al, 2017; Deverman et al, 2016).

AAV vectors were delivered to mice retro-orbitally at p28 at $10^{11}$ vg per animal as previously described (Chan et al, 2017). Animals were anesthetized using 0.01 ml/g Avertin (1.3% 2,2,2-Tribromoethanol (T48402-25G) and 0.7% 2-methyl-2-butanol (Sigma 240486) in Phosphate Buffered Saline (PBS)) prior to injection. Animals were monitored post-injection, and tissues were harvested at timepoints indicated.

## Immunoblotting

Mouse cortical tissues were sonicated in RIPA buffer containing 1x protease inhibitor cocktail (Sigma, P8340) and spun down at 21,000 $\times g$ for 20 min at 4 °C. Protein levels were standardized across all samples using a BCA Protein Assay kit (Thermo Fisher, 23,225) and 4× Laemmli buffer was added to samples to a final concentration of 1×. Cultured cells were lysed directly in 1× Laemmli buffer, sonicated, and spun down at 21,000 $\times g$ for 10 min at Room temperature (RT).

Samples were boiled and run on 7.5% or 10% polyacrylamide gels in tris/glycine buffer before transferring to polyvinylidene difluoride (PVDF) membranes. PVDF membranes were blocked with 5% milk in tris-buffered saline containing 0.3% Tween-20 (TBST) or Everyblot blocking buffer (Bio-Rad Laboratories,12010020) for 1–2 h at RT before applying primary antibodies overnight at 4 °C. After washing with TBST, the membranes were incubated with the appropriate horseradish peroxidase (HRP) or fluorescent protein-conjugated secondary antibodies for 1 h at RT. Membranes were washed again with TBST, developed using enhanced chemiluminescence substrate when appropriate (Kindle Biosciences, R1004), and imaged using a ChemiDoc Imager (Bio-Rad) or KwikQuant Imager (Kindle Biosciences). Blots were analyzed using ImageJ software (NIH). See Appendix Table S2 for antibody information.

## Human samples

ALS and nonneurological control paraffin-embedded postmortem motor cortex sections were obtained from the JHU ALS Postmortem Resources Core. For all human samples used in this study, procedures were performed under protocols approved by the Institutional Review Board (IRB) of Johns Hopkins University. The demographics of postmortem samples used in this study are provided in Appendix Table S1.

## Immunohistochemistry (IHC)

Mice were anesthetized with 0.02 ml/g Avertin solution (1.3% 2,2,2-Tribromoethanol (T48402-25G) and 0.7% 2-methyl-2-butanol (Sigma 240486) in Phosphate Buffered Saline (PBS)) before transcardial perfusion with 0.1 M Phosphate Buffer (PB) and 4% Paraformaldehyde (PFA) in 0.1 M PB. Brains were dissected, post-fixed in 4% PFA for 18-20 h, washed with PBS, and prepared for embedding in cryomolds or paraffin blocks as previously described (Cave et al, 2017).

Cryo-embedded brains were sectioned on a cryostat (Leica CM3050 or Leica CM3050 S) at 30–40 μm. IHC-immunofluorescence (IF) was performed on free-floating sections. Paraffin blocks were cut into 4-μm sections on a Rotary microtome (Leica RM2235) and collected on slides. Paraffin sections were deparaffinized with Xylenes and rehydrated in an ethanol series immediately prior to staining. For tissues from mice older than 12 months old and used for IHC-IF, autofluorescence quenching was performed for 3 h at 4 °C (Sun et al, 2017b).

Sections were washed in PBS and permeabilized in 0.3% Triton-X-100 in PBS (PBST). For paraffin sections, antigen retrieval was performed with 0.1 M sodium citrate buffer (pH 6) for 10 min in a microwave or 20 min in a 95 °C water bath. Sections were then incubated for at least 1 h with blocking solution (5% Normal Donkey Serum (NDS) or 5% Bovine Serum Albumin (BSA) (Sigma, A9647-100G) in PBS and 0.3% Triton-X-100 in PBS). Sections were incubated overnight with primary antibodies at 4 °C and washed with PBS the next day. For DAB staining, endogenous peroxidases were blocked by washing with 0.3% $H_2O_2$ in PBS two times for 15 min each prior to adding secondary antibodies. Sections were then incubated with the appropriate fluorescently or HRP-conjugated secondary antibodies (Jackson Immunoresearch) for 1–2 h at RT. For IF, nuclei were stained with Hoechst 33342 (1:500, Thermo Fisher) in PBS for 15 min at RT. Sections were mounted on slides with Prolong Gold mounting media (Thermo Fisher, P36931) and coverslipped before imaging. Images were acquired with a Zeiss LSM700 microscope. The same settings were used for all images acquired within the same experiment.

For DAB staining, slides were washed again with PBS, and 3,3′-Diaminobenzidine (DAB) solution (Sigma-Aldrich, D4168-50) was applied to sections for visualization. Slides were then washed with dH₂O, mounted with Prolong Gold mounting media (Thermo Fisher, P36931), and coverslipped before imaging. Brightfield images were taken with a Keyence BZ-X710 epifluorescence microscope. Identical contrast and brightness adjustments were made across experimental groups.

Human tissue sections were deparaffinized and rehydrated as described above. Autofluorescence quenching was performed overnight at 4 °C (Sun et al, 2017b), and IHC-DAB was performed as described above, followed by incubation with a second primary antibody overnight at 4 °C to allow for co-staining. After washing with PBS, an alkaline phosphatase-conjugated secondary antibody (Jackson Immunoresearch, 1:500) was applied to sections for 1 h at RT. Slides were washed with PBS, and tissue sections were stained with ImmPACT Vector Red Alkaline Phosphatase Substrate Kit (Vector laboratories, SK-5105) according to the manufacturer's instructions. Sections were then washed with PBS and coverslipped with Prolong Diamond (Invitrogen, P36970) before imaging. A Keyence BZ-X710 brightfield/epifluorescence microscope was used for image acquisition. Contrast and brightness adjustments were made equally across experimental groups. See Appendix Table S2 for antibody information.

## Immunocytochemistry (ICC)

Cells were grown on acid-washed coverslips for ICC. Primary cortical neuronal cultures were fixed with 4% PFA at 4 °C for 10 min, and iSN cultures were fixed with 4% PFA at RT for 20 min. Fixed cells were permeabilized with PBST and blocked with 5%

NDS in PBST for at least 1 h. Cells were then incubated at 4 °C with primary antibody overnight, followed by washes with PBS. Cells were incubated with the appropriate fluorescently conjugated secondary antibodies for 1 h at room temperature. Nuclei were stained with Hoechst 33342 (1:500, Thermo Fisher) in PBS for 15 min at RT, followed by washes with PBS. Coverslips were then mounted on Superfrost Plus microscope slides in ProLong Gold Antifade Mountant and imaged on Zeiss LSM700 confocal microscope. See Appendix Table S2 for antibody information.

## Transmission electron microscopy (TEM)

Animals were perfused with 2% glutaraldehyde 2% paraformaldehyde (freshly prepared form EM grade prill) 50 mM sodium cacodylate 50 mM phosphate (Sorenson's) 5 mM $MgCl_2$, pH 7.4 at 1085 mOsmols, and kept at 4 °C for 2 h. Whole brains were removed and placed in fresh fixative overnight at 4 °C. The following steps were kept cold (4 °C) until the 70% ethanol step, then run at RT. Samples were rinsed in 50 mM cacodylate 50 mM phosphate 3% sucrose 5 mM $MgCl_2$, pH 7.4 at 430 mOsmols for 45 min, and then microwaved (50% power, 10 s pulse-20 s pause-10 s pulse, Pelco laboratory grade microwave model 3400), post-fixed in 2% osmium tetroxide reduced with 1.6% potassium ferrocyanide, in the same buffer without sucrose then placed on ice in the dark for 2 h. Samples were then rinsed in 100 mM maleate buffer with 3.5% sucrose pH 6.2, then en-bloc stained for 1 h with filtered 2% uranyl acetate in maleate sucrose buffer, pH 6.2. Following en-bloc staining, samples were dehydrated through a graded series of ethanol to 100%, transferred through propylene oxide, embedded in Eponate 12 (Pella), and cured at 60 °C for 2 days.

Sections were cut on a Reichert Ultracut E microtome with a Diatome Diamond knife (45 degree). In all, 60-nm sections were picked up on formvar coated $1 \times 2$ mm copper slot grids and stained with methanolic uranyl acetate followed by lead citrate. Grids were viewed on a Hitachi 7600 TEM operating at 80 kV, and digital images were captured with an XR80 8-megapixel CCD by AMT.

### NPC quantification

TEM studies analyzed deep-layer neurons of the parietal cortex, identified by size and morphology. Images focused on the nuclear envelope, which was acquired at 80,000× magnification. A minimum of 290 NPCs across 30 cells from three different mice per genotype were analyzed. NPC width was determined by measuring the length of a line drawn from one inner edge of the NPC to the other in ImageJ (NIH). The average was compiled and compared between groups using Student's *t* test.

## Quantitative image analysis

All images were quantified using ImageJ software (NIH). For mouse tissues, 2–6 sections per mouse were analyzed depending on the experiment. For human tissues, 10–12 regions of interest (ROIs) per section were chosen at random to be imaged and analyzed. For in vitro experiments, at least 50 MAP2+ neurons were analyzed. All manual quantification was performed blinded.

### Chromogenic stains

TDP-43+ cells were manually quantified from DAB-stained mouse tissues. Cells were counted as TDP-43+ if they contained staining throughout the cell body and the signal was not below a relative intensity threshold set to WT in ImageJ. For GDE2 and TDP-43 quantification from human sections, cells were scored for GDE2 accumulations as in Westerhaus et al, 2022. Briefly, cells with accumulations of GDE2 were manually quantified from paraffin sections of control ($n = 5$) and ALS patient ($n = 6$) motor cortices. For each sample, 16 regions of interest (0.212 mm²) per section were chosen at random to be imaged and analyzed. DAB staining (brown) for GDE2 was considered an accumulation if it was above a relative intensity threshold set to WT using ImageJ. In cells with GDE2 accumulations, TDP-43 abnormalities were scored in two categories: TDP-43 loss and TDP-43 cytoplasmic mislocalization. TDP-43 loss was visually scored as the absence of TDP-43 (red) staining. Cytoplasmic mislocalization was visually scored as the presence of TDP-43 (red) in the cytoplasm with reduction or loss in the nucleus. This was confirmed by examining corresponding immunofluorescence images of TDP-43 alkaline phosphatase staining.

### Fluorescent stains

Neurons with Nup98 puncta were manually quantified. Puncta were identified based on their relative intensity compared to background. Threshold-based cell counting macros created in ImageJ were used to quantify NeuN or MAP2-positive cells and cells double positive for both NeuN or MAP2 and TDP-43, Ctip2, or MYC. Neurons were not considered positive for TDP-43 if their intensity was more than 1.5 standard deviations below the mean WT intensity after correcting for background intensity differences. Area fractions were calculated in ImageJ as the area covered by the respective marker over the total area measured.

To quantify nuclear/cytoplasmic (N/C) or C/nuclear envelope (NE) ratios, DAPI or Lamin B was used to delineate the nucleus from the cytoplasm in neurons. NeuN signal outside of DAPI/Lamin B was used to define the cytoplasm. The nuclear and cytoplasmic intensities were measured as the average fluorescence intensities from three ROIs of the same size in the nucleus and in the cytoplasm in the channel of interest. The NE intensity was measured by drawing a freehand line of 2-pixel width around the nucleus. Five background measurements were taken per image, the average of which was used to correct for differences in background intensity. The final N/C ratio was calculated by dividing the average background-corrected nuclear intensity by the average background-corrected cytoplasmic intensity for each cell. The final C/NE ratio was calculated by dividing the average background-corrected cytoplasmic intensity by the average background-corrected NE intensity for each cell. Nuclear intensities were calculated using a mask of the area within DAPI to measure mean intensity in the relevant channel of interest.

## Cell culture

Mouse primary cortical cultures were prepared from P0 or P1 mouse cortices and plated on poly-l-lysine-coated plates or acid-washed coverslips as previously described (Nakamura et al, 2021) with minor modifications. Cells are maintained in Neurobasal medium supplemented with 2% B27, 1% l-glutamine, and 1% Pen/Strep. In total, 5 μM cytosine arabinoside was added on DIV2 to inhibit glial growth and removed on DIV3. From DIV4, cultures were fed every 3 days and maintained at 37 °C until harvest at DIV21.

Control and *C9orf72* iPSC lines were obtained from the Answer ALS repository at Cedars-Sinai Biomanufacturing Center (Appendix Table S3). All experiments in iPSCs were carried out under protocols approved by the IRB of Johns Hopkins University under the study number IRB00355223. Feeder-free iPSCs were maintained in mTeSR plus (StemCell Technologies) on growth factor reduced Matrigel (Corning). iPSCs were fed daily and split every 5–7 days using EZ-LiFT StemCell Passaging Reagent (Sigma-Aldrich) according to the manufacturer's instructions. Differentiation of iPSCs into spinal neurons was carried out according to the direct induced motor neuron protocol as previously described (Coyne et al, 2020; Workman et al, 2023) (https://neurolincs.org/pdf/diMN-protocol.pdf).

For GSK3-β inhibitor experiments, iSN cultures of control lines were treated with DMSO, 5 or 10 µM SB216763 starting on day 25 and harvested on day 32. For Wnt inhibitor experiments, iSN cultures of ALS patient lines were treated with DMSO, 0.5, 1, or 2 µM LGK974 starting on day 39 and collected on day 46. Differentiations were carried out in batches of 4 lines (2 control, 2 patient lines). Each experiment was performed with four control lines and four patient lines, each with at least two independent rounds of differentiation.

## RNA extraction, RT-PCR, and qRT-PCR

For mouse cortical tissues, total RNA was isolated using RNeasy Mini or Midi kit (Qiagen). In total, 1 µg of total RNA was used to synthesize cDNA using the SuperScript® III First-Strand Synthesis System (Thermo Fisher) with random primers. RT-PCR detecting *Camk1g, Tecpr1,* and *Synj2bp* cryptic exon was performed using previously described primers and touch-down PCR protocol (Korbie and Mattick, 2008).

The total RNA was isolated from day 32 and day 46 iSN cultures for qRT-PCR analysis using Trizol (Invitrogen) according to the manufacturer's instructions. In total, 600–2000 ng of total RNA was used to synthesize cDNA using the High-Capacity cDNA Reverse Transcription Kit (Thermo Fisher). qPCR was then performed using Fast SYBR Green Master Mix (Thermo Fisher) on the StepOne plus Real-Time PCR system (Applied Biosystems) using the $\Delta\Delta$Ct method according to the manufacturer's instructions. mRNA abundance was measured relative to *GAPDH* mRNA. Technical duplicates or triplicates were utilized within each qRT-PCR plate. See Appendix Table S4 for qRT-PCR and RT-PCR primer sequences.

## RNAseq and analysis

### Extraction, library preparation, and HiSeq sequencing
RNA extraction, library preparations, and sequencing reactions were conducted at GENEWIZ, LLC. (South Plainfield, NJ, USA) as follows:

The total RNA was extracted using Qiagen RNeasy Plus Universal mini kit following the manufacturer's instructions (Qiagen, Hilden, Germany). Extracted RNA samples were quantified using Qubit 2.0 Fluorometer (Life Technologies, Carlsbad, CA, USA), and RNA integrity was checked using Agilent TapeStation 4200 (Agilent Technologies, Palo Alto, CA, USA). All samples had a RIN ≥8.4.

RNA sequencing libraries were prepared using the NEBNext Ultra II RNA Library Prep Kit for Illumina following the manufacturer's instructions (NEB, Ipswich, MA, USA). Briefly, mRNAs were first enriched with Oligo(dT) beads. Enriched mRNAs were fragmented for 15 min at 94 °C. First-strand and second-strand cDNAs were subsequently synthesized. cDNA fragments were end-repaired and adenylated at 3'ends, and universal adapters were ligated to cDNA fragments, followed by index addition and library enrichment by limited-cycle PCR. The sequencing libraries were validated on the Agilent TapeStation (Agilent Technologies, Palo Alto, CA, USA), and quantified by using Qubit 2.0 Fluorometer (Invitrogen, Carlsbad, CA) as well as by quantitative PCR (KAPA Biosystems, Wilmington, MA, USA).

The sequencing libraries were clustered on two flowcell lanes. After clustering, the flowcell was loaded on the Illumina HiSeq instrument 4000 according to the manufacturer's instructions. The samples were sequenced using a 2 × 150 bp Paired-End (PE) configuration. On average, approximately 70 million reads from 4-month cortices and 50 million reads from 19-month cortices were obtained. Image analysis and base calling were conducted by the HiSeq Control Software (HCS). Raw sequence data (.bcl files) generated from Illumina HiSeq was converted into fastq files and de-multiplexed using Illumina's bcl2fastq 2.17 software. One mismatch was allowed for index sequence identification.

## Data analysis

Raw sequencing reads were processed for differential expression and differential splicing analysis.

For differential splicing, sequence reads were trimmed to remove possible adapter sequences and nucleotides with poor quality using Trimmomatic v.0.36. The trimmed reads were mapped to the *Mus musculus* reference genome available on ENSEMBL using the STAR aligner v.2.5.2b. BAM files were generated as a result of this step and used as input for splicing analysis using Leafcutter (Li et al, 2018). Differential splicing of intron clusters was tested using Chi-squared test. Intron clusters with a FDR < 0.1 and $|\Delta PSI| > 0.1$ were identified as significantly differentially spliced clusters. The sequences of these clusters were then retrieved from GRCm38, available on ENSEMBL, and used as input for Multiple Em for Motif Elicitation (MEME) (Bailey and Elkan, 1994) analysis using the MEME Suite. A 1st order model of sequences built on the input sequences was used as the background, and motif sites were restricted to the input strand only as RNA. A motif containing TG repeats was recovered and then used as input for Find Input Motif Occurrences (FIMO) (Grant et al, 2011) analysis to look for instances of the motif in the input intron sequences and test for significant enrichment.

For differential gene expression analysis, sequence reads were mapped to the *Mus musculus* reference transcriptome GRCm38 and quantified using Kallisto (Bray et al, 2016). Sleuth (Pimentel et al, 2017) was then used for downstream differential expression analysis by fitting the data to a linear model of genotypes. The Wald test was used to test the effect of genotypes, generate *P* values, effect sizes (estimated changes associated with each model parameter, b) and standard error of effect sizes (se_b). *b/se_b* was used as the input rank for non-parametric gene set enrichment analysis (GSEA) using the fgsea (Gennady et al, 2021) package. Genes with

Benjamini–Hochberg (BH) adjusted *P* values (FDR) < 0.1 and effect size b > 0.5 were called as significantly differentially expressed genes, and pathways with BH adjusted *P* values (FDR) < 0.05 and absolute enrichment score >1 were labeled differential pathways.

## Statistical analysis

Data were analyzed and plotted using GraphPad Prism (version 9). Statistical significance for pairwise comparisons was derived using a two-tailed Student's *t* test with relevant corrections. For multiple comparisons, we used ANOVA with corrections for multiple comparisons. All values are reported as mean ± s.e.m. In figures, asterisks denote statistical significance: $*P < 0.05$, $**P < 0.01$, $***P < 0.001$, $****P < 0.0001$. Specific statistical information for each experiment is included in the figure legends.

# Data availability

All sequencing data generated and analyzed for this study (Datasets EV1 and EV2) have been deposited to the Gene Expression Omnibus under the accession number GSE246462. This paper does not report the original code. Differential gene expression and differential splicing analysis were carried out following manuals for Kallisto/Sleuth and Leafcutter, respectively. The source data for this study is provided with the manuscript. Microscopy images used for quantification have been deposited to the BioImage Study Archive under the accession number S-BIAD1153. Correspondence and requests for materials should be addressed to Dr. Shanthini Sockanathan at ssockan1@jhmi.edu.

The source data of this paper are collected in the following database record: biostudies:S-SCDT-10_1038-S44318-024-00156-8.

# Peer review information

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

## Acknowledgements

The authors thank Y Li and D Sama-Borbon for technical assistance; Dr. J Nathans for *Wnt-MYC-GFP* mice; Dr. M Taketo (Kyoto University) for *Ctnnb*^*flex3* mice; Dr. V. Gradinaru (CalTech) for AAV-PHP.eB rep-cap plasmids; Dr. J Lin for bioinformatics support; Irika Sinha for help with cryptic exon detection by RT-PCR; Dr. J Xu and Dr. B Zaepfel for help with iPSCs; the Johns Hopkins ALS Postmortem Tissue Core for patient samples; the Department of Neuroscience Multiphoton Imaging core, Mike Delannoy for assistance with electron microscopy (Johns Hopkins Microscope Facility) and the Developmental Studies Hybridoma Bank for the MYC (9E10) and NKX6-1 antibody. AAV-GFP/ Cre was a gift from Dr. Fred Gage; Lentiviral-S-tdTomato was a gift from Dr. Jeffrey Rothstein. This work was funded by grants to AW (NIH grant T32GM007445, NIH F31AG072745) and SS (NIH 5RO1AG068043).

## Author contributions

**Nan Zhang**: Conceptualization; Data curation; Formal analysis; Validation; Investigation; Visualization; Methodology; Writing—original draft; Writing— review and editing. **Anna Westerhaus**: Conceptualization; Data curation; Formal analysis; Funding acquisition; Validation; Investigation; Visualization; Methodology; Writing—original draft; Writing—review and editing. **Macey Wilson**: Formal analysis; Validation; Visualization. **Ethan Wang**: Data curation; Formal analysis; Visualization. **Loyal Goff**: Formal analysis; Supervision. **Shanthini Sockanathan**: Conceptualization; Formal analysis; Supervision; Funding acquisition; Visualization; Methodology; Writing—original draft; Project administration; Writing—review and editing.

Source data underlying figure panels in this paper may have individual authorship assigned. Where available, figure panel/source data authorship is listed in the following database record: biostudies:S-SCDT-10_1038-S44318-024-00156-8.

## Disclosure and competing interests statement

The authors declare no competing interests.

# Expanded View Figures

**Figure EV1.   GDE2 loss leads to neurodegeneration and neuronal loss.**

(**A–H**) Representative images of immunohistochemical staining of cortical sections of 19-month animals. (**d'–d'''**) highlight changes in microglial morphology in *Gde2*KOs. Scale bar: (**A–H**) = 50μm; (**c'–d'''**) = 15 μm. (**I–L**) Graphs quantifying area fraction for astrogliosis (**I**, GFAP, *P = 0.0147), microglial activation (**J**, Iba1, *P = 0.0127), neuronal numbers (**K**, NeuN, *P = 0.0495), and deep-layer neurons (**L**, Ctip2, *P = 0.0180). All graphs: mean ± s.e.m., Unpaired *t* test. *n* = 5-11 WT, 5-11 *Gde2*KO.

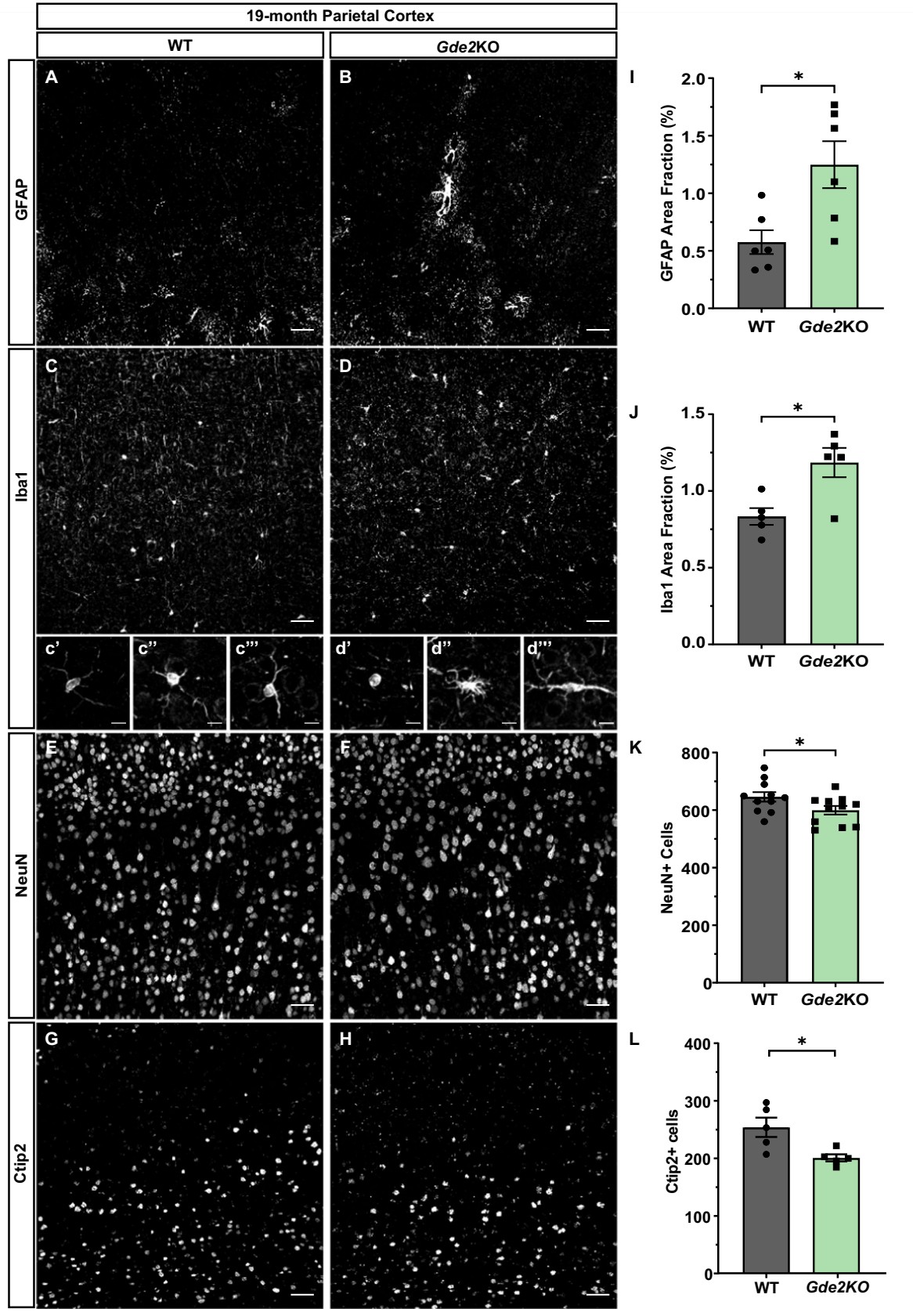

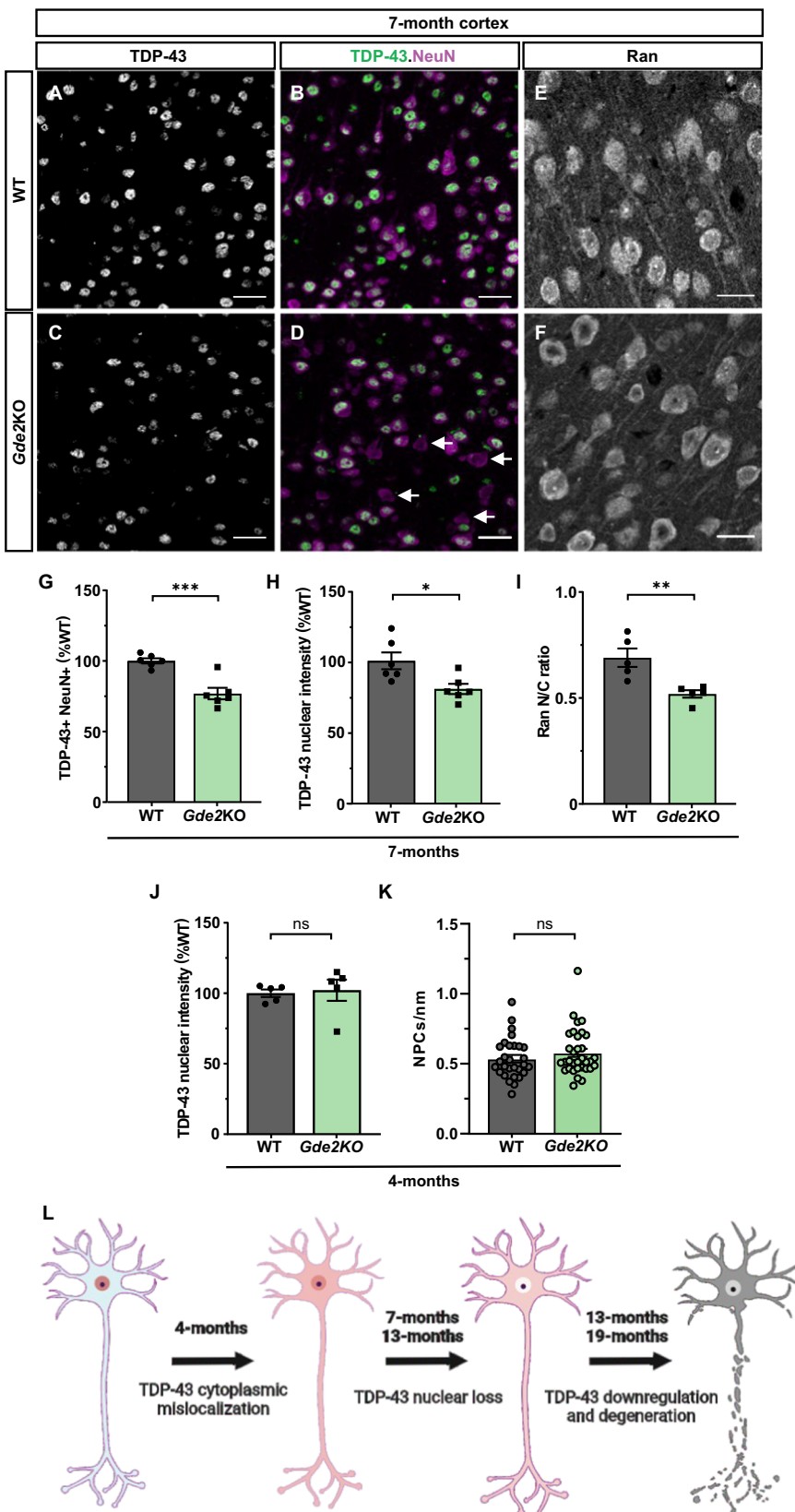

**◀**

**Figure EV2.  Progression of TDP-43 expression and NPC number in WT and *Gde2*KO animals.**

(A–F) Representative images of immunohistochemical staining of cortical sections of 7-month WT and *Gde2*KO animals. Arrows (D) highlight neurons lacking TDP-43 expression. Scale bar: (A–F) = 25 µm. (G–I) Graphs quantifying the percentage of neurons expressing TDP-43 (G, ***$P = 0.0004$), neuronal TDP-43 nuclear intensity (H, *$P = 0.0173$), and Ran N/C ratio (I, **$P = 0.0066$) in 7-month old animals. $n = 6$ WT, 6 *Gde2*KO. (J) Graph quantifying neuronal TDP-43 nuclear intensity in 4-month animals (J, ns $P = 0.7977$). $n = 5$ WT, 5 *Gde2*KO. (K) Graph quantifying NPCs per nm from WT and *Gde2*KO TEM micrographs (ns $P = 0.2859$). $n = 29$ WT cells, 33 *Gde2*KO cells, 3 animals/genotype. (L) Schematic describing the progression of TDP-43 expression over time in *Gde2*KO mice. All graphs: mean ± s.e.m., Unpaired *t* test. Schematic in (L) created with BioRender.com

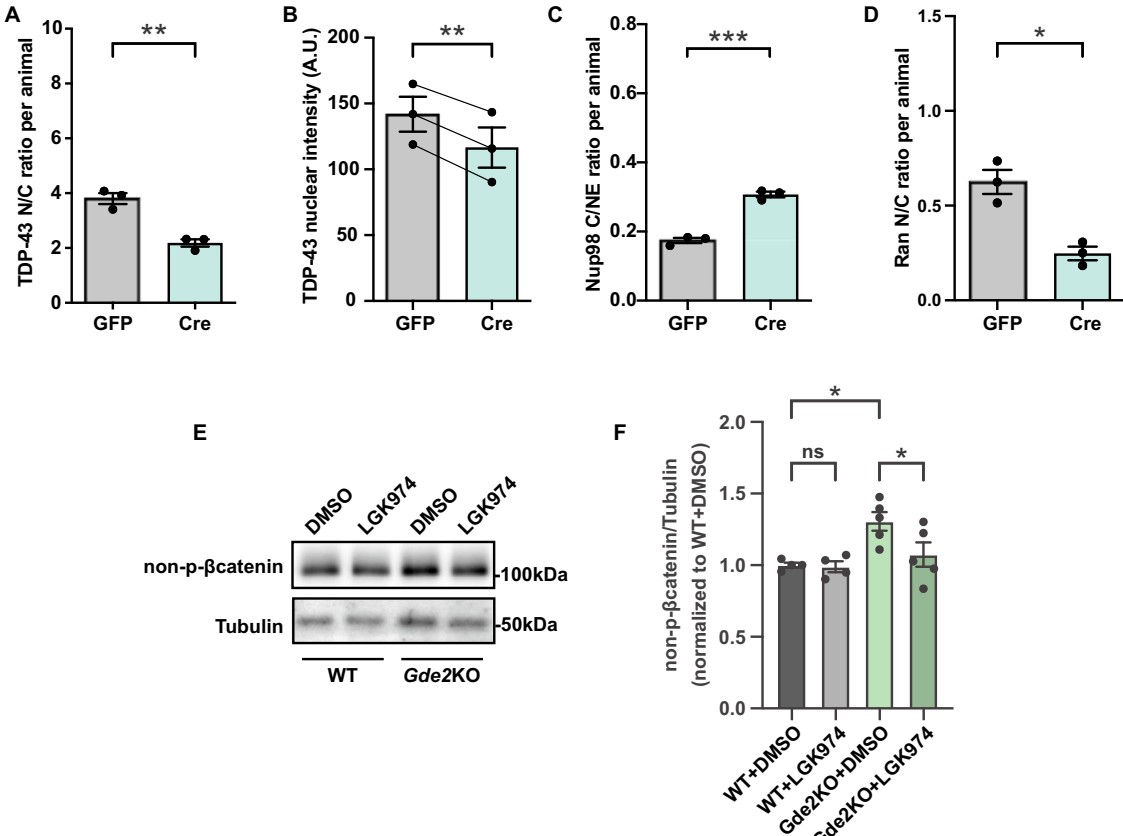

**Figure EV3. Stabilized endogenous β-catenin causes TDP-43 nuclear exclusion, changes Nup98 distribution, and disrupts NCT.**

(A–D) Presentation of data from Fig. 4M–P comparing the number of animals analyzed. *Ctnnb^flex3* mice were injected with AAV expressing GFP or Cre.GFP. Graphs quantifying neuronal TDP-43 nuclear/cytoplasmic ratio (**A**, \**P* = 0.0041), TDP-43 nuclear intensity (**B**, \**P* = 0.0066; A.U.= arbitrary units), Nup98 cytoplasmic/nuclear envelope ratio (**C**, \*\**P* = 0.0003) and Ran nuclear/cytoplasmic ratio (**D**, \**P* = 0.0125). *n* = 3 animals per condition. (**E**) Representative Western blot of primary cortical neurons prepared from WT and *Gde2*KO mice treated with vehicle (DMSO) or LGK974. (**F**) Graph quantifying non-phosphorylated β-catenin in WT and *Gde2*KO cultured cortical neurons treated with DMSO or LGK974 (WT + DMSO/WT + LGK974) ns = 0.9993, (WT + DMSO/*Gde2*KO + DMSO) \**P* = 0.0128, (*Gde2*KO + DMSO/ *Gde2*KO + LGK974) \**P* = 0.0472. *n* = 4 WT and 5 *Gde2*KO cultures. All graphs: mean ± s.e.m., (**A, C, D**): Welch's *t* test; (**B**): paired *t* test. (**F**) One-way ANOVA with Šidák's multiple comparisons test.

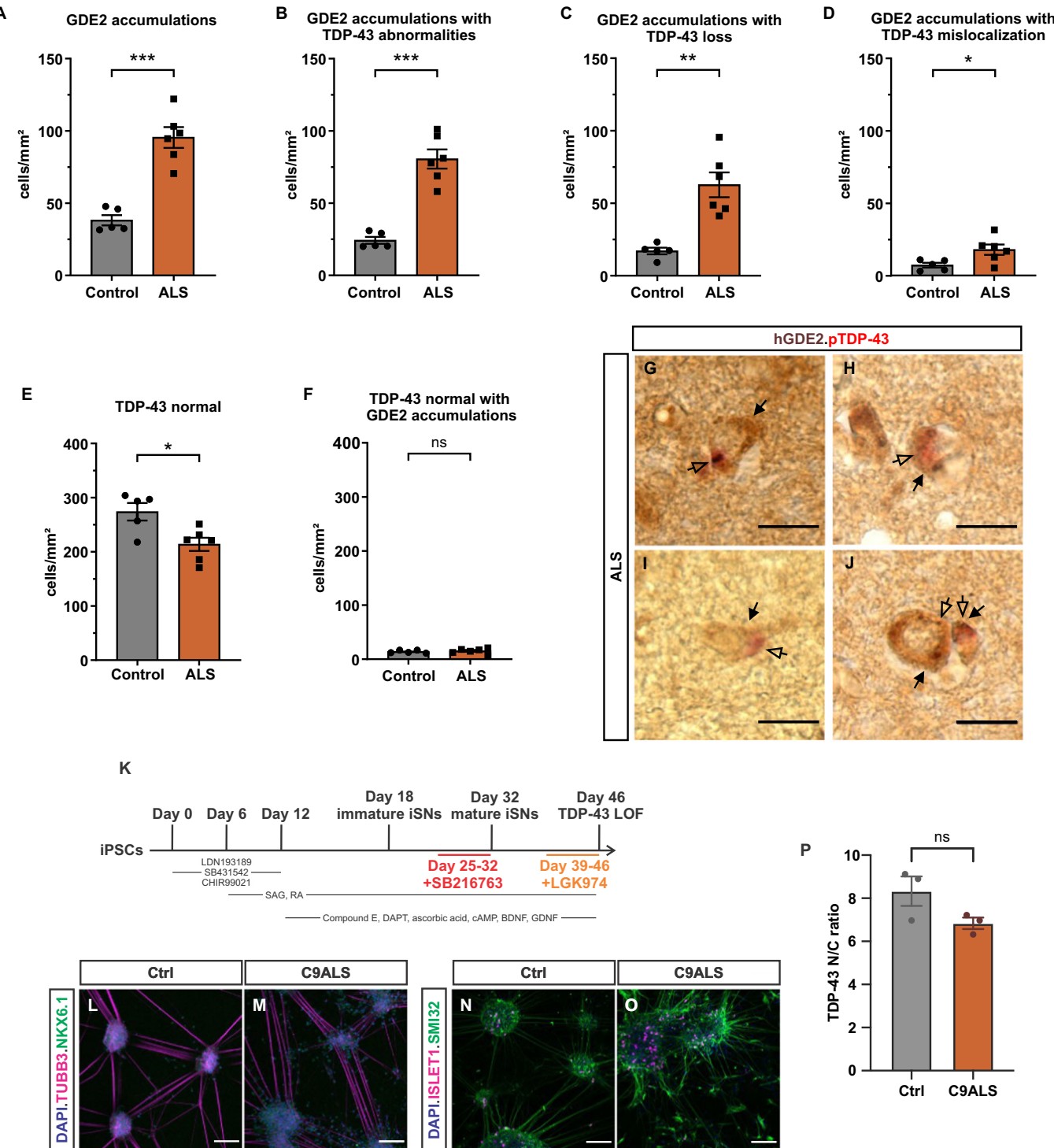

**Figure EV4.  Analysis of GDE2 and TDP-43 expression in postmortem brain of control and patients with ALS, and differentiation of iSNs.**

(A–F) Graphs showing comparisons between control individuals and patients with ALS from compiled graphs in Fig. 5A,B. (A) ***$P$ = 0.0002; (B) ***$P$ = 0.0002; (C) **$P$ = 0.0025; (D) *$P$ = 0.0322; (E) *$P$ = 0.0144; (F) ns $P$ = 0.6542. $n$ = 5 controls, $n$ = 6 patients with ALS. (G–J) Representative images of immunohistochemical staining from postmortem ALS patient brains showing exemplar cells with hGDE2 accumulations (arrows) that overlap with pTDP-43 inclusions (open arrows). Scale bar = 15 µm. (K) Schematic detailing the differentiation timeline of iSNs from iPSCs. (L–O) Representative images of immunocytochemical staining of Day 32 iSNs for the motor neuron and dorsal interneuron marker ISLET1, the progenitor marker Nkx6.1, tubulin (TUBB3), and neurofilament (SMI32). Scale bar = 100 µm. (P) Graph comparing TDP-43 N/C ratios in Day 46 Ctrl and C9ALS iSNs. ns $P$ = 0.1122, $n$ = 3. All graphs: mean ± s.e.m., (A–C) Welch's $t$ test; (D–F, P) Unpaired $t$ test.

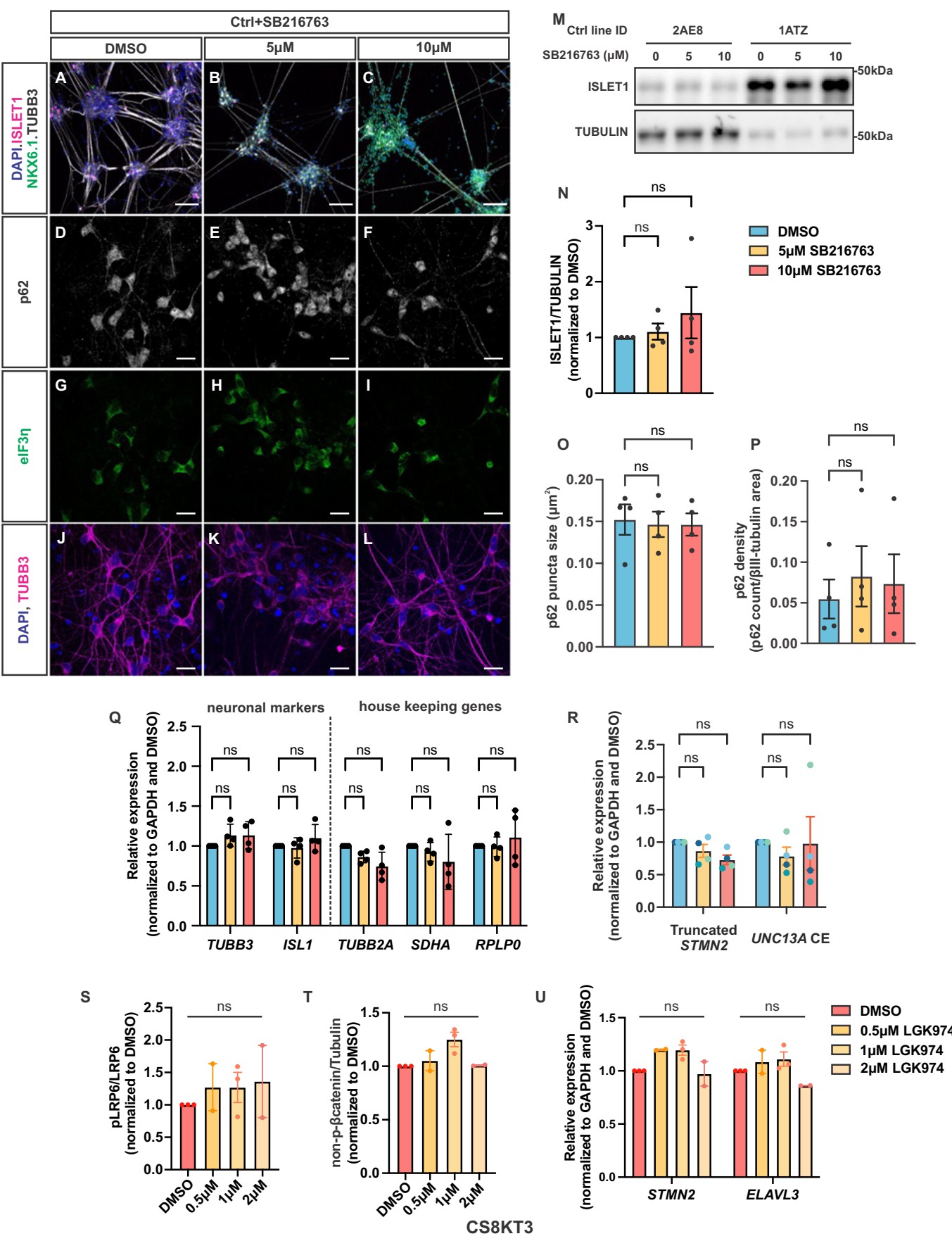

◀ **Figure EV5. Wnt activation and inhibition of iSNs.**

(A–L) Representative images of immunocytochemical staining of Day 32 Ctrl iSNs treated with DMSO or SB216763 for ISLET1, Nkx6.1, tubulin (TUBB3), neurofilament (SMI32), stress granules (eIF3η) and autophagosome (p62). (M) Representative western blot of iSN protein extracts from Day 32 Ctrl lines treated with different concentrations of the GSK3-β inhibitor, SB216763, for 1 week. (N) Graph quantifying ISLET1 protein normalized to tubulin when treated with DMSO and 5 µM (ns $P > 0.9999$) or 10 µM of SB216763 (ns $P > 0.9999$), $n = 4$ cell lines. (O, P) Graphs quantifying autophagosome (p62) size (O) and density (P) in Ctrl iSNs treated with DMSO or SB216763. p62 size (O) (DMSO/5 µM) ns $P = 0.8940$, (DMSO/10 µM) ns $P = 0.8782$; density (P) (DMSO/5 µM) ns $P = 0.2495$, (DMSO/10 µM) ns $P = 0.4258$, $n = 4$ cell lines. (Q) Graphs quantifying mRNA level of motor neuron markers and other housekeeping genes in Ctrl iSNs treated with DMSO, 5 µM or 10 µM SB216763. *TUBB3* (DMSO/5 µM) ns $P = 0.4068$, (DMSO/10 µM) ns $P = 0.4001$; *ISL1* (DMSO/5 µM) ns $P = 0.9687$, (DMSO/10 µM) ns $P = 0.5995$; *TUBB2A* (DMSO/5 µM) ns $P = 0.3850$, (DMSO/10 µM) ns $P = 0.0584$; *SDHA* (DMSO/5 µM) ns $P = 0.7469$, (DMSO/10 µM) ns $P = 0.1609$; *RPLP0* (DMSO/5 µM) ns $P = 0.9944$, (DMSO/10 µM) ns $P = 0.5546$, $n = 4$ cell lines, 2 differentiations/line. (R) Graph quantifying cryptic exon inclusion of *STMN2* (Truncated *STMN2*) and *UNC13A* (*UNC13A* CE) in Ctrl iSNs treated with DMSO, 5 µM or 10 µM SB216763. Truncated *STMN2* (DMSO/5 µM) ns $P = 0.8259$, (DMSO/10 µM) ns $P = 0.4878$; *UNC13A* CE (DMSO/5 µM) ns $P = 0.6283$, (DMSO/10 µM) ns $P = 0.9963$ $n = 4$ cell lines, 2 differentiations/line. (S, T) Graphs quantifying phosphorylated LRP6 normalized to total LRP6 protein (DMSO/0.5 µM) ns $P > 0.9999$, (DMSO/1 µM) ns $P > 0.9999$, (DMSO/2 µM) ns $P > 0.9999$ (S), and non-phosphorylated β-catenin normalized to tubulin (DMSO/0.5 µM) ns $P > 0.9999$, (DMSO/1 µM) ns $P = 0.0609$, (DMSO/2 µM) ns $P > 0.9999$ (T) in C9ALS line CS8KT3 treated with different concentrations of the Wnt inhibitor, LGK974. $n = 2$ or 3 differentiations. (U) Graphs quantifying mRNA level of TDP-43 target gene expression in C9ALS line CS8KT3 treated with different concentrations of the Wnt inhibitor, LGK974. *STMN2* (DMSO/0.5 µM) ns $P = 0.0843$, (DMSO/1 µM) ns $P = 0.0535$, (DMSO/2 µM) ns $P = 0.9806$; *ELAVL3* (DMSO/0.5 µM) ns $P = 0.6698$, (DMSO/1 µM) ns $P = 0.3824$, (DMSO/2 µM) ns $P = 0.2875$. $n = 2$ or 3 differentiations. All graphs: mean ± s.e.m. (N) Friedmen's test with Dunn's multiple comparisons test; (O, P) RM one-way ANOVA, with Geisser-Greenhouse correction and Dunnett's multiple comparisons test; (Q, R, U) Two-way ANOVA with Dunnett's multiple comparisons test (Q, R) or Šidák's multiple comparisons test (U). (S, T) Kruskal–Wallis test with Dunn's multiple comparisons test. $n = 4$ cell lines. Scale bar (A–C) = 100 µm; (D–L) = 20 µm.

