## [Peer Review File · The EMBO Journal]

Physiological regulation of neuronal Wnt activity is essential for TDP-43 localization and function

Nan Zhang, Anna Westerhaus, Macey Wilson, Ethan Wang, Loyal Goff, and Shanthini Sockanathan

Corresponding author: Shanthini Sockanathan (ssockan1@jhmi.edu)

Review Timeline:

Submission Date:	16th Jan 24
Editorial Decision:	13th Feb 24
Revision Received:	30th Apr 24
Editorial Decision:	29th May 24
Revision Received:	31st May 24
Accepted:	10th Jun 24

Editor: Kelly Anderson

Transaction Report:

Dear Prof. Sockanathan,

Thank you for submitting your manuscript for consideration by the EMBO Journal. It has now been seen by two referees whose comments are shown below.

Given the referees' positive recommendations, I would like to invite you to submit a revised version of the manuscript, addressing the comments of both reviewers. I should add that it is EMBO Journal policy to allow only a single round of revision, and acceptance of your manuscript will therefore depend on the completeness of your responses in this revised version. It would be good to discuss your plan to address the referee concerns and I am available to do so in the coming weeks by zoom or email.

Thank you for the opportunity to consider your work for publication. I look forward to your revision.

Yours sincerely,

Kelly M Anderson, PhD
Editor, The EMBO Journal
k.anderson@embojournal.org

We realize that it is difficult to revise to a specific deadline. In the interest of protecting the conceptual advance provided by the work, we recommend a revision within 3 months (13th May 2024). Please discuss the revision progress ahead of this time with the editor if you require more time to complete the revisions.

Referee #1:

The author present a compelling paper on the role of Gde2 in maintaining TDP-43 nuclear localization mediated by Wnt/beta-catenin signal. The work is relevant for the ALS field and well performed. The study is carried out mainly in animal model and then some finding are confirmed in iPSCs.

I found the first part very compelling and there is huge potential for the animal model to be used to study sporadic ALS; the authors although need to address some concerns aimed at proving the validity of the model for the study of ALS.

The second part on hiPSCs is less technically sounded and more issues need to be addressed.

The authors also failed in explaining why the Gde2-Wnt-beta-cat axis is involved in NPC, NCT, TDP-43 localization. From their data is evident that the two events (Gde2KO and TDP-43 nuclear loss/NPC disruption) have a correlation but there is no mechanistic explanation. I understand though that this question might be the topic for another paper and thus I think the present paper is fit for a publication in EMBO J after the major revisions listed below.

Major:

- Given the potentiality of this model to be a model for sporadic ALS it would be interesting to measure the behavioral phenotype of this mice. From figure one the thinning of the cortex is very evident and I would be surprised if it does not correlate with a cognitive phenotype (cognitive and motor see point below)
- From source publication of the mouse model (DOI 10.1016/j.neuron.2011.07.028) it looks like GDE2 KO affected motor neurons maturation. The authors should include evaluation of TDP-43 localization in motor neurons as well and assessment of behavior.
- Does LGK974 decrease b-cat? Does ablation of b-cat in Gde2KO mice rescue the phenotype? These experiments will give a more complete picture of the mechanism.
- Figure 5: do inclusions contain pTDP-43?
- Figure 6 A and C: total levels of TDP-43 by WB are not sufficient to state that there is not TDP-43 abnormalities. Immunofluorescence for TDP-43 should be performed and N/C should be assessed for every tested concentration of SB216763
- Figure 6D: TDP-43 targets decrease can be a sign of TDP-43 loss of function but also can be a sign of neuronal distress, while appearance of STMN2CE or UNC13A are a most accepted sign of TDP-43 loss of function. The presence of CE should be tested as well by qPCR.
- Figure 6E. only C9line are presented, while controls should be presented as well especially when a rescue is claimed. TDP43 nuclear and cytoplasmic localization should be tested for the treatment as well in both controls and C9-ALS iPSN.

Referee #2:

In this manuscript, Zhang et al. propose a new context for TDP-43 mislocalization in which glycerophosphodiester phosphodiesterase 2 (GDE2) regulates the activity of canonical Wnt signaling, which results in cytoplasmic accumulation of TDP-43. GDE2 knockout mice developed an age-progressive accumulation of TDP-43, progressing to nuclear loss of TDP-43 at 7 months, and degeneration of cortical neurons at 13 and 19 months of age. Mice lacking Gde2 (Gde2KO) showed deficits in nucleocytoplasmic transport, as characterized by decreased nuclear localization of Ras-related GTPase (Ran). Transcriptomic analysis of Gde2KO mice identified Wnt signaling as differentially regulated. To determine if sustained Wnt signaling is sufficient to induce TDP-43 pathology, the authors utilized a mouse model with stabilized B-catenin and suggest there is mislocalization of Ran, Nup98 and TDP-43. The authors further propose GDE2 abnormalities coincide with TDP-43 mislocalization in post-mortem ALS patient tissue. Similarly, they observe increased Wnt activation and decreased GDE2 in induced human spinal neuron (iSN) cultures derived from ALS patients. Overall, the authors present some compelling data and use several model systems which will be of interest to the neurodegeneration and neuroscience community. However, several of their claims are not supported by the data provided and the manuscript would be strengthened by addressing the following comments.

Major concerns:

-In Fig. 1G, the stated presence of a cryptic exon in Camk1g is not clear. The band indicated by the arrow around 500 bp is only clearly present in 1 of 3 Gde2 KO lanes, and also is present in the first and third lanes of the WT cortex. This does not seem to match the 3 points in the quantification in 1H where 2 Gde2KO points appear nearly the same and only one is about the same level as WT.

Fig2 I-J and Fig.4 F and H, why does the Nup98 IHC look different? Figure 4F and H shows bright band around nucleus, that is not apparent in previous figure. Is there a mix-up with the images in Figure 2?

The authors state that "4-month Gde2KO animals showed an increase in Nup98 cytoplasmic aggregates compared with WT, accompanied by a decrease in Nup98 staining, (Figure 2I, J O)". However, there is no data that these are cytoplasmic aggregates. The overall staining is more dim in Gde2 KO, with some small areas of slightly brighter intensity, but no experiments were done to determine that these were in fact protein aggregates. The authors should revise their interpretation.

In Fig4. the Ran IHC with LGK974 appears to be less intense overall. It is not evident from the image provided that there is an increase in nuclear Ran and decrease in cytosolic. Perhaps the overall decrease in cytoplasmic Ran is skewing the ratio and nuclear Ran is not changed.

Fig S2. Iba1 should stain all microglia and is not a marker of activation per se. Microglial activation could be inferred based on microglial morphology but the images too low resolution to observe. It looks like IHC for Iba1 was not successful in WT mice as it should stain all microglia and there are patches with no staining.

Figure S3. The figure title states that GDE2 expression is required for neuronal survival. However, the NeuN numbers do not change between groups WT and Gde2KO mice. There is no evidence provided that loss of Gde2KO induces neurodegeneration.

Fig 5. I do not see other cells in image provided that meet the criteria for having both GDE2 accumulations and lost nuclear TDP-43. It is not clear how the authors reached the quantification in C and D. Based on how the quantification is plotted, it is also not clear whether there is variability between patients. Is this an average cells/mm² across all 5 controls and 6 ALS?

Minor concerns:

The authors state that LGK974 did not affect Myc expression in non-neuronal cells. That is surprising considering it is a small molecule. How were non-neuronal cells identified?

For most of the analyses, error bars should be plotted as SD and not SEM. It is more informative to the reader to understand variability between samples and not proximity to the mean.

In Figure 6D, are the points technical replicates or biological replicates (ie. independent experiments)? How many times was this experiment repeated?

Response to Reviews

Overall, the referees were very positive, with Referee #1 stating, “The author present a compelling paper on the role of *Gde2* in maintaining TDP-43 nuclear localization mediated by Wnt/beta-catenin signal. The work is relevant for the ALS field and well performed.... there is huge potential for the animal model to be used to study sporadic ALS” and Referee #2 stating, “Overall, the authors present some compelling data and use several model systems which will be of interest to the neurodegeneration and neuroscience community”. However, both referees had concerns, which we address below.

Referee #1:

Major:

- Given the potentiality of this model to be a model for sporadic ALS it would be interesting to measure the behavioral phenotype of this mice. From figure one the thinning of the cortex is very evident and I would be surprised if it doe snot correlate with a cognitive phenotype (cognitive and motor see point below)

We recently published a manuscript describing cognitive phenotypes in *Gde2*KO mice (Daudelin et al., [DOI.org/10.1186/s12993-024-00234-1](https://doi.org/10.1186/s12993-024-00234-1)). The behavioral outcomes are referred to in the revised Introduction.

- From source publication of the mouse model (DOI 10.1016/j.neuron.2011.07.028) it looks like *GDE2* KO affected motor neurons maturation. The authors should include evaluation of TDP-43 localization in motor neurons as well and assessment of behavior.

We have qualitative data showing that TDP-43 in 2-month old *Gde2*KO animals is mislocalized in spinal motor neurons. We have included this in the following figure for the reviewer’s perusal. However, we are unable to generate quantitative data to include in the manuscript as this would require generating new animals, which would take several months and is outside the window for resubmission.

Reviewer Figure 1. TDP43 is mislocalized in the spinal cord of *Gde2*KO animals. A-h'''. Sections of spinal cord from 2-month-old animals. ChAT expression identifies motor neurons. Panels e'-f''' show predominantly nuclear TDP-43 expression in ChAT+ neurons in WT spinal cord, with occasional motor neurons showing cytoplasmic TDP-43 expression (e'''). However, almost all ChAT+ motor neurons in

*Gde2*KO spinal cords show expression of TDP-43 in the nucleus and cytoplasm (arrows in **G and H**, panels **g'-h''**). Scale bar **A-H**:50 μ m; **e'-h'''**: 15 μ m.

The assessment of motor behavior in *Gde2*KO mice has already been published (*Cave et al.*, 2017 DOI: [10.1186/s13024-017-0148-1](https://doi.org/10.1186/s13024-017-0148-1)). These animals show hindlimb claspings, reductions in grip strength and fine motor movement, as well as increased heat sensitization consistent with altered motor and sensory function. We have referred to this study in the revised introduction.

- Does LGK974 decrease *b-cat*? Does ablation of *b-cat* in *Gde2*KO mice rescue the phenotype? These experiments will give a more complete picture of the mechanism.

We show in **Fig.6I** that LGK974 treatment lowers β -catenin amounts in human iSNs generated from C9ALS patients. In mice, LGK974 treatment of *Gde2*KOs dampens neuronal Wnt signaling but not the robust Wnt signaling in non-neuronal cells. Given this confound, evaluating the effects of LGK974 on non-phosphorylated β -catenin by Western blot using whole tissue lysates would not be informative. Accordingly, we have treated primary mouse WT and *Gde2*KO neuronal cultures with LGK974. We include new data in **Fig. EV3E and F** showing that LGK974 reduces the amounts of non-phosphorylated β -catenin in cultured *Gde2*KO neurons, confirming that it decreases Wnt activation.

Genetic KO of β -catenin leads to gastrulation defects at the blastula stage of embryogenesis (DOI: [10.1083/jcb.148.3.567](https://doi.org/10.1083/jcb.148.3.567); DOI: [10.1242/dev.121.11.3529](https://doi.org/10.1242/dev.121.11.3529)); thus, conventional KOs of β -catenin cannot be used to ablate β -catenin in *Gde2*KOs. β -catenin has other essential roles besides mediating Wnt-signaling, such as in cellular adhesion (doi.org/10.1098/rsob.200267); therefore, even neuron-specific conditional KO of β -catenin is likely to have pleiotropic effects, complicating interpretation. Taken together, ablating β -catenin in *Gde2*KO mice would not be informative for understanding roles in Wnt-dependent phenotypes.

- Figure 5: do inclusions contain pTDP-43?

We have stained sections of postmortem brain of patients with ALS using our GDE2 antibody and published antibodies against pTDP-43. We find that cells with GDE2 accumulations contain pTDP-43, and in some cases, GDE2 accumulations coincide with pTDP-43 expression. We have included this data in revised **Fig. EV4G-J**.

- Figure 6 A and C: total levels of TDP-43 by WB are not sufficient to state that there is not TDP-43 abnormalities. Immunofluorescence for TDP-43 should be performed and N/C should be assessed for every tested concentration of SB216763.

We performed this experiment in iSNs, as suggested by the reviewer, and found that TDP-43 nuclear expression is reduced in response to SB216763 treatment. We have included these data in revised **Fig. 6E and F**.

- Figure 6D: TDP-43 targets decrease can be a sign of TDP-43 loss of function but also can be a sign of neuronal distress, while appearance of *STMN2CE* or *UNC13A* are a most accepted sign of TDP-43 loss of function. The presence of CE should be tested as well by qPCR.

The reduction of TDP-43 target transcripts and the incorporation of cryptic exons (CE) in *STMN2* and *UNC13A* transcripts are typically observed in iSNs under conditions where TDP-43 function is

completely abolished/knocked down or in patient tissues (DOI:[10.1038/s41593-018-0293-z](https://doi.org/10.1038/s41593-018-0293-z); DOI:[10.1038/s41593-018-0300-4](https://doi.org/10.1038/s41593-018-0300-4); DOI: [10.1038/s41586-022-04424-7](https://doi.org/10.1038/s41586-022-04424-7); DOI: [10.1038/s41586-022-04436-3](https://doi.org/10.1038/s41586-022-04436-3)).

However, **CE inclusion is not consistently observed in iSNs in patient lines**; instead, the **downregulation of TDP-43 targets is reproducibly detected in iSNs when TDP-43 activity is decreased** (DOI: [10.1016/j.celrep.2023.113046](https://doi.org/10.1016/j.celrep.2023.113046); DOI: [10.1038/s41593-018-0300-4](https://doi.org/10.1038/s41593-018-0300-4); DOI: [10.1126/scitranslmed.abe1923](https://doi.org/10.1126/scitranslmed.abe1923)). This is evident in published studies using the same differentiation protocol as our study, where TDP-43 localization and expression either show modest changes or appear unchanged (DOI: [10.1126/scitranslmed.abe1923](https://doi.org/10.1126/scitranslmed.abe1923)). Indeed, recent RNA-seq studies detect no evidence of transcripts with CEs in patient iSNs but identify reductions in TDP-43 target RNAs (DOI:[10.1016/j.celrep.2023.113046](https://doi.org/10.1016/j.celrep.2023.113046)). We have included new data in **Fig. EV5R** showing that we do not detect CEs in iSNs treated with SB216763. The apparent discrepancies between iSNs and patient tissue where CEs are detected may be due to impaired RNA clearance mechanisms in disease. Accordingly, the examination of target gene levels is a more reproducible method to detect TDP-43 loss of function when TDP-43 expression is not completely ablated and RNA clearance mechanisms are intact.

Should neurons be in such distress that TDP-43 target transcripts are reduced through TDP-43 independent mechanisms (such as impaired transcription), then we would expect a global reduction of cellular transcript levels. We show that the transcription of a panel of genes encompassing structural, metabolic, ribosomal, and motor neuron transcripts is not compromised by the addition of the Wnt agonist, arguing against global effects on transcription due to neuronal distress (**Fig. EV5Q**). In support of this notion, we include new data in revised **Fig. EV5A-P**, showing that neuronal identity, stress granule formation, and autophagy are not altered by treating neurons with SB216763.

- Figure 6E. only C9line are presented, while controls should be presented as well especially when a rescue is claimed. TDP43 nuclear and cytoplasmic localization should be tested for the treatment as well in both controls and C9-ALS iPSN.

We have presented the results this way because we are comparing the C9 lines treated with either vehicle alone or with the Wnt inhibitor. The control lines were not treated with vehicle or inhibitor and thus are not appropriate for comparative analysis. We also excluded one C9 cell line we initially examined for TDP-43 molecular dysfunction because it did not exhibit Wnt suppression in response to the drug, making it difficult to pair the two experiments for comparative analysis. We have now changed the text to state “increased expression” or “partial rescue”.

Published studies of C9 lines using the same differentiation protocol and using some of the lines we employ in our study show that changes in TDP-43 localization are typically not observed in C9ALS iSNs at Day 46 (the time point of our study) (DOI: [10.1126/scitranslmed.abe1923](https://doi.org/10.1126/scitranslmed.abe1923)). We have confirmed this to be the case and include this data in Fig. S8P.

Referee #2:

Major concerns:

-In Fig. 1G, the stated presence of a cryptic exon in Camk1g is not clear. The band indicated by the arrow around 500 bp is only clearly present in 1 of 3 Gde2 KO lanes, and also is present in the first and third lanes of the WT cortex. This does not seem to match the 3 points in the quantification in 1H where 2 Gde2KO points appear nearly the same and only one is about the same level as WT.

When quantified, the data stands as shown with an increase in *Camk1g* cryptic exon (CE) in the *Gde2*KOs. To address the reviewer’s concern, we have examined two other targets of TDP-43 function (see revised **Fig.1G-I**). The increase in CE is clearly shown for *Synj2bp*, where the appearance of the fragment relates to CE inclusion. We note that there is some variation in CE inclusion in particular genes

amongst animals; however, all *Gde2*KOs show CE inclusion spread across different genes, reflective of TDP43 dysfunction. This variation may reflect the stochastic nature of TDP-43 downregulation in *Gde2*KOs that could affect different neuronal types in different animals. We have noted this in the revised text.

Fig2 I-J and Fig.4 F and H, why does the Nup98 IHC look different? Figure 4F and H shows bright band around nucleus, that is not apparent in previous figure. Is there a mix-up with the images in Figure 2?

There is no mix-up with the images. In **Fig 2I-J**, the samples were prepared from paraffin sections of 4-month-old tissues, while in **Fig 4F and H**, the samples were analyzed using frozen (cryo) sections 3 weeks post AAV injection as a means to capture early changes upon Wnt activation. These different preparations and different time points result in different IHC appearances for Nup98. However, we note that the band around the nucleus is still observed in **Fig. 2I** for the WT condition.

The authors state that "4-month Gde2KO animals showed an increase in Nup98 cytoplasmic aggregates compared with WT, accompanied by a decrease in Nup98 staining, (Figure 2I, J O)". However, there is no data that these are cytoplasmic aggregates. The overall staining is more dim in Gde2 KO, with some small areas of slightly brighter intensity, but no experiments were done to determine that these were in fact protein aggregates. The authors should revise their interpretation.

We thank the reviewer for pointing this out, and we have modified the text to state "puncta" rather than "aggregates".

In Fig4. the Ran IHC with LGK974 appears to be less intense overall. It is not evident from the image provided that there is an increase in nuclear Ran and decrease in cytosolic. Perhaps the overall decrease in cytoplasmic Ran is skewing the ratio and nuclear Ran is not changed.

We thank the reviewer for pointing this out, and we have changed the image to better reflect the quantified data. We note, however, that the Ran gradient is critical for nuclear-cytoplasmic transport and that measuring the nuclear/cytoplasmic ratio is the most accepted method for measuring this gradient.

Fig S2. Iba1 should stain all microglia and is not a marker of activation per se. Microglial activation could be inferred based on microglial morphology but the images too low resolution to observe. It looks like IHC for Iba1 was not successful in WT mice as it should stain all microglia and there are patches with no staining.

We have included better images for Iba1 staining and also higher magnification images to provide clarity on microglial morphology. We have also changed the text to address the reviewer's concern

Figure S3. The figure title states that GDE2 expression is required for neuronal survival. However, the NeuN numbers do not change between groups WT and Gde2KO mice. There is no evidence provided that loss of Gde2KO induces neurodegeneration.

We have changed the Figure title to say that "GDE2 expression in the adult is required to prevent neurodegenerative changes". We note that **Appendix Fig. S2** relates to an earlier timepoint (13 months) when there is no neuronal loss in *Gde2*KOs; however, there is a neuronal loss in 19-month *Gde2*KO animals (**Fig. EV1**) consistent with neurodegeneration.

Fig 5. I do not see other cells in image provided that meet the criteria for having both GDE2 accumulations and lost nuclear TDP-43. It is not clear how the authors reached the quantification in C

and D. Based on how the quantification is plotted, it is also not clear whether there is variability between patients. Is this an average cells/mm² across all 5 controls and 6 ALS?

Similar to quantification in Westerhaus et al., 2022 (DOI: 10.1186/s40478-022-01376-x), cells with accumulations of GDE2 were manually quantified from paraffin sections of control (n=5) and ALS patient (n=6) motor cortices. For each sample, 16 regions of interest (0.212 mm²) per section were chosen at random to be imaged and analyzed. DAB staining (brown) for GDE2 was considered an accumulation if it was above a relative intensity threshold set to WT using ImageJ. In cells with GDE2 accumulations, TDP-43 abnormalities were scored in two categories: TDP-43 loss and TDP-43 cytoplasmic mislocalization. TDP-43 loss was visually scored as the absence of TDP-43 (red) staining. Cytoplasmic mislocalization was visually scored as the presence of TDP-43 (red) in the cytoplasm with reduction or loss in the nucleus. This was confirmed by examining corresponding immunofluorescence images of TDP-43 alkaline phosphatase staining. This has been clarified in the methods section, and additional example images of cells with GDE2 accumulations and TDP-43 abnormalities are included in revised **Fig. 5A and B** to provide clarity.

Panel C refers to the total cells with GDE2 accumulations, and of these cells, the number of cells with both GDE2 accumulations and TDP-43 abnormalities (nuclear loss or cytoplasmic mislocalization) were plotted. Panel D refers to the total TDP-43+ cells per image, and of these cells, TDP-43+ cells, which also had GDE2 accumulations, were plotted. These are plotted as the average of cells/mm² by patient of control and ALS, respectively. The variability between patients for these analyses is shown in **Fig. EV4**, with each dot representing an individual patient average.

Minor concerns:

The authors state that LGK974 did not affect Myc expression in non-neuronal cells. That is surprising considering it is a small molecule. How were non-neuronal cells identified?

Non-neuronal cells were identified as Myc+NeuN- (ie lacking NeuN expression). It is possible that the non-neuronal cells may respond to treatments of higher concentrations of LGK974.

For most of the analyses, error bars should be plotted as SD and not SEM. It is more informative to the reader to understand variability between samples and not proximity to the mean.

We respectfully point out that error bars plotted as SEM are widely used in published literature and in the EMBO journal.

In Figure 6D, are the points technical replicates or biological replicates (ie. independent experiments)? How many times was this experiment repeated?

The points refer to 4 different iPSC lines, each subject to 2 separate differentiations. This is stated in the Figure legend.

Dear Shanthini,

Congratulations on a great revision! Overall, the referees have been positive, however there remain a few editorial items that I ask you to attend to in a revised manuscript. When you submit your revised version, please address the following editorial items and add this to a new point-by-point response:

1. Please provide up to five keywords, which may or may not appear in the title, should be given in alphabetical order, below the abstract, each separated by a slash (/).
2. Please move the references to before the figure legends.
3. Please move the data availability section before the acknowledgments section.
4. Please rename the conflict of interest section to "Disclosure Statement and Competing Interests".
5. Please remove the author contribution section from the main manuscript.
6. We require that all figures be referred to in the main manuscript and in chronological order, please add a callout to figure 2K, L, M, N, 3C, and 4T.
7. For the dataset EVs, there are two zipped datasets; each should be unzipped and uploaded as a Dataset and each legend should be provided in its Excel file as a separate sheet/tab.
8. For the appendix file, the table of contents should have listed each Appendix item with a page number; "Supplementary Table S4" should be changed to "Appendix Table S4".
9. Please rename the material and methods section to "Methods".
10. In the figure legends, please ensure that figures 3c-k and 4s-x are provided sequentially.
11. Please provide the exact p value in the legend of figures 2r, 4m-p, s.
12. Please describe the nature of N in the legends of figures 1e, l-m.
13. Please provide a scale bar for figures 3a, a-b', f-g, i-j; 4a-c, e-g, i-k, q, u, w; EV 4g-i, l, n; EV 5a-k.
14. Please provide a scale bar and its definition for figures 1a, c, j; 2a, a', c, c', c", d', e-g, e'-g', i; 5a'-a', b'-b'; EV 1a-g, c'-d'; EV 2a-c, e.

Thank you for the opportunity to consider your work for publication and I look forward to your revision.

Warm wishes,
Kelly

Kelly M Anderson, PhD
Editor, The EMBO Journal
k.anderson@embojournal.org

Further information is available in our Guide For Authors: <https://www.embopress.org/page/journal/14602075/>

authorguide

Referee #1:

the manuscript is now in a ready-to-publish format

Referee #2:

The authors have sufficiently addressed this reviewer's concerns. This work will be of interest to the neurodegeneration and neuroscience communities.

The authors addressed the minor formatting issues.

Dear Shan,

Congratulations on an excellent manuscript, I am pleased to inform you that your manuscript has been accepted for publication in the EMBO Journal. Thank you for your comprehensive response to the referee concerns and for providing detailed source data. It has been a pleasure to work with you to get this to the acceptance stage.

I will begin the final checks on your manuscript before submitting to the publisher next week. Once at the publisher, it will take about 3 weeks for your manuscript to be published online. As a reminder, the entire review process including referee concerns and your point-by-point response will be available to readers.

I will be in touch throughout the final editorial process until publication. In the meantime, I hope you find time to celebrate!

Warm wishes,
Kelly

Kelly M Anderson, PhD
Editor, The EMBO Journal
k.anderson@embojournal.org
